# Microbial occurrence and symbiont detection in a global sample of lichen metagenomes

**Gulnara Tagirdzhanova[1]¤, Paul Saary[2], Ellen S. Cameron[2,3], Carmen C. G. Allen[1], Arkadiy I. Garber[4], David Díaz Escandón[1], Andrew T. Cook[1], Spencer Goyette[1,5], Veera Tuovinen Nogerius[6], Alfredo Passo[7], Helmut Mayrhofer[8], Håkon Holien[9], Tor Tønsberg[10], Lisa Y. Stein[1], Robert D. Finn[2]\*, Toby Spribille[1]\***

**1** Department of Biological Sciences, University of Alberta, Edmonton, Canada, **2** European Molecular Biology Laboratory, European Bioinformatics Institute (EMBL-EBI); Hinxton, United Kingdom, **3** Wellcome Sanger Institute; Hinxton, United Kingdom, **4** Biodesign Center for Mechanisms of Evolution and School of Life Sciences, Arizona State University; Tempe, Arizona, United States of America, **5** University of British Columbia Herbarium, University of British Columbia, Vancouver, Canada, **6** Department of Ecology and Genetics, Uppsala University; Uppsala, Sweden, **7** Instituto de Investigaciones en Biodiversidad y Medioambiente, CONICET—Universidad Nacional de Comahue, Bariloche, Argentina, **8** Institute of Biology, University of Graz, Graz, Austria, **9** Faculty of Biosciences and Aquaculture, Nord University, Steinkjer, Norway, **10** Department of Natural History, University Museum of Bergen, University of Bergen, Bergen, Norway

¤ Current address: The Sainsbury Laboratory, University of East Anglia; Norwich, United Kingdom
\* rdf@ebi.ac.uk (RDF); toby.spribille@ualberta.ca (TS)

**Data Availability Statement:** The sequencing data in this project are submitted to ENA: de novo generated raw data (study accession PRJEB59037), metagenomic assemblies

## Abstract

In lichen research, metagenomes are increasingly being used for evaluating symbiont composition and metabolic potential, but the overall content and limitations of these metagenomes have not been assessed. We reassembled over 400 publicly available metagenomes, generated metagenome-assembled genomes (MAGs), constructed phylogenomic trees, and mapped MAG occurrence and frequency across the data set. Ninety-seven percent of the 1,000 recovered MAGs were bacterial or the fungal symbiont that provides most cellular mass. Our mapping of recovered MAGs provides the most detailed survey to date of bacteria in lichens and shows that 4 family-level lineages from 2 phyla accounted for as many bacterial occurrences in lichens as all other 71 families from 16 phyla combined. Annotation of highly complete bacterial, fungal, and algal MAGs reveals functional profiles that suggest interdigitated vitamin prototrophies and auxotrophies, with most lichen fungi auxotrophic for biotin, most bacteria auxotrophic for thiamine and the few annotated algae with partial or complete pathways for both, suggesting a novel dimension of microbial cross-feeding in lichen symbioses. Contrary to longstanding hypotheses, we found no annotations consistent with nitrogen fixation in bacteria other than known cyanobacterial symbionts. Core lichen symbionts such as algae were recovered as MAGs in only a fraction of the lichen symbioses in which they are known to occur. However, the presence of these and other microbes could be detected at high frequency using small subunit rRNA analysis, including in many lichens in which they are not otherwise recognized to occur. The rate of MAG recovery correlates with sequencing depth, but is almost certainly influenced by biological attributes of organisms that affect the likelihood of DNA extraction, sequencing and successful assembly, including cellular abundance, ploidy and strain co-occurrence. Our results suggest that,

(PRJEB72384, PRJEB72386-PRJEB72404, PRJEB72498- PRJEB72501), and MAGs (PRJEB77567). Phylogenomic trees in Newick format are available at FigShare (10.6084/m9. figshare.27054937). Custom scripts used for data analysis and visualization are available on GitHub (https://github.com/Spribille-lab/2024-Microbial-occurrence-in-lichen-metagenomes) and FigShare (10.6084/m9.figshare.27054937).

**Funding:** Support for this study was provided by funding to GT from an Alberta Graduate Excellence Scholarship and Alberta Innovates Graduate Student Scholarship. CCGA was supported by NSERC PGS-D grant NSERC PGS-D fellowship 545691-2020. GT, DDE, SG, ATC, VTN and TS were supported by Natural Sciences and Engineering Research Council of Canada (NSERC) Discovery Grant RGPIN-2019-04892 to TS, and funding to TS through the Canada Research Chairs (CRC) Program. PS, ESC and RDF were supported by European Molecular Biology Laboratory (EMBL) funds. The funders were not involved in conceptualization, study design, data collection, analysis, manuscript preparation or decision to publish.

**Competing interests:** The authors have declared that no competing interests exist.

**Abbreviations:** AAP, aerobic anoxygenic phototroph; ANI, average nucleotide identity; CBB, Calvin–Benson–Bassham; GH, glycoside hydrolases; GTDB, Genome Taxonomy Database; KO, KEGG ortholog; LFS, lichen fungal symbiont; MAG, metagenome-assembled genome; SSU, small subunit.

though metagenomes are a powerful tool for surveying microbial occurrence, they are of limited use in assessing absence, and their interpretation should be guided by an awareness of the interacting effects of microbial community complexity and sequencing depth.

## Introduction

In some biological systems, unrelated organisms have evolved interactions so stable and integrated that they appear to function, to the casual observer, as one. This phenomenon was discovered over 150 years ago in lichens [1,2]. Long thought to constitute a single organism, lichens were revealed to be a tightly integrated relationship of a fungus and phototrophic alga and/or cyanobacterium (collectively often called "photobionts"), in which fungi are thought to receive sugar alcohols or glucose from the photobiont [3,4]. Subsequent research has revealed the pairings currently classified under the umbrella of "lichens" to have arisen multiple times in both fungal and photobiont evolution [5,6], to engage in different schemes of metabolic exchange [7–9], and to involve, to varying degrees, additional constitutively associated organisms [10–14].

All symbioses currently classified as lichens conform to a few basic patterns. All are ectotrophic [15] and all symbionts—including algae [16]—are partially or wholly osmotrophs. Participating microbes exchange goods and services across shared cell wall contacts and mucilages [17], making lichens essentially highly structured biofilms [18]. The lichen fungal symbiont (LFS), which is not known to produce a free-living vegetative life stage in nature and never reproduces outside of symbiosis [4,9], has been shown to produce, together with its mucilage, about an order of magnitude more biomass than the photobiont [17]. LFS species likewise far outnumber those of the photobiont. Given their asymmetrical biomass and diversity, the LFS is typically accorded the role that in other symbiotic systems has been called the host. Most eukaryotic photobionts, by contrast, are regularly detected in the free-living state in surrounding environment, reproduce almost exclusively outside of the lichen symbiosis [9], and like cyanobacterial photobionts are widely shared among LFSs [19]. Although much about the biology of the disparate photobiont lineages remains unclear, especially with respect to what they receive from the fungus, their population structuring vis-à-vis their LFS partners is consistent with the definition of an open symbiosis [20].

The only universally recognized organismal associate of lichen fungi, and central to countless definitions of "lichen" [21,22], is the photobiont, the presence of which is determined today, no different than 150 years ago, by visual inspection under a microscope. No consensus exists however on how many other symbionts might be involved, and since most other associated microbes lack visible chlorophyll, their detection is not trivial. Researchers as early as Maria Cengia Sambo in the 1920s cultured bacteria from lichens, leading to the speculation that lichens could be "polysymbioses" of more than 2 partners [23]. Culture-based screening of lichens in the search for functional bacterial contributors to lichen symbioses continued into the 21st century [24,25], and the detection of associated strains and potentially shared metabolic products led Grube and Berg [10] to begin referring to these microbes as "bacteriobionts." However, though hundreds of bacterial strains have been detected, overall few lichen symbioses have been censused for their bacterial composition, with most work focused on the model symbiosis involving the ascomycete fungus *Lobaria pulmonaria* (reviewed by Grimm and colleagues [13]). Similarly, early culture-based detections of basidiomycete yeasts from lichens have been followed by metatranscriptome studies and PCR screening suggesting high

frequency occurrence, especially in lichens involving fungi from the family Parmeliaceae [12]. All of these approaches have been limited in their ability to capture the totality of lichen microbial composition by the inherent biases of microscopic detectability, culturability, the use of specific primers, or, in the case of metatranscriptomes used to study basidiomycete yeasts, selection of eukaryotic mRNA via poly-A tailing [12].

The shotgun-based DNA sequencing approach underlying metagenomics would appear to offer a solution to surveying total organismal composition in lichens owing to its theoretical ability to conduct an unbiased census of genomes, much as it has in other experimentally recalcitrant biological systems [26]. Hundreds of lichen metagenomes have been published in the last decade in the context of the study of lichen microbial composition [25–32] and recovery and analysis of the LFS genome [33–37], and these metagenomes have collectively begun to be mined for their protein diversity [38,39]. The recent use of lichen metagenomes to argue both for [14,27,29,30,31,32] and against [28,32] specific organismal compositions of lichen samples has raised questions about the overall organismal information content of currently available lichen metagenomes, their uses and their limitations. In particular, since no broad-scale survey of lichen metagenome content has been undertaken, underlying assumptions regarding their ability to capture basic lichen organismal composition, such as known symbionts, have not been tested.

In the present study, we reassembled and analyzed 456 lichen metagenomes to produce metagenome-assembled genomes (MAGs) and facilitate the mapping of the overall organismal composition, in a geographically wide sample of lichen symbioses drawn from 12 different studies. We complemented these with 24 additional, newly generated metagenomes that had been sequenced more deeply than the average publicly available data. We mapped organismal composition of broad taxonomic groups based on individual MAGs as well as detection of unambiguous rRNA in assemblies and raw reads. We gave special emphasis to testing for recovery of the canonically recognized photobionts, as well as putative basidiomycete symbionts and high-frequency bacterial lineages. We also annotated and compared the occurrence of core functional biosynthetic pathways in highly complete prokaryotic and eukaryotic MAGs and explored potential cross-feeding patterns in the small number of metagenomes in which multiple complete MAGs were recovered together.

## Results

### Metagenomic assembly and binning yield 1,000 MAGs

We started with a data set of 456 lichen metagenomes from 12 different published sources [14,27,28,30–37,40] (Fig 1 and S1 Table), plus 24 metagenomes generated de novo (S2 Table). This data set included representatives of most major lichen groups (S1 Fig). Although the majority of the metagenomes were generated from samples collected in North America, 6 continents were represented in total (S2 Fig). We began by removing identical metagenomes (not biological replicates) that have been published twice in the sequence databases (*n* = 43; S3 Table). Next, we assembled and binned each metagenome individually, obtaining 17,390 bins (Fig 1A, see Methods). We obtained 1,142 MAGs above the quality threshold QS50 [39] (hereafter "QS50 MAGs"). Since different genotypes of the same organism can occur in multiple samples, we dereplicated the MAG set to obtain species-level representatives, defined at 95% average nucleotide identity (ANI; hereafter "species").

The final set included 1,000 QS50 MAGs: 674 bacterial, 294 fungal, and 32 algal (S4 Table). For each of these taxonomic groups, we constructed phylogenomic trees (Fig 2) based on sets of genes shared across all MAGs found in the same taxonomic same taxonomic group, thereby contextualizing the relationship of the MAGs to each other and to reference genomes (listed in

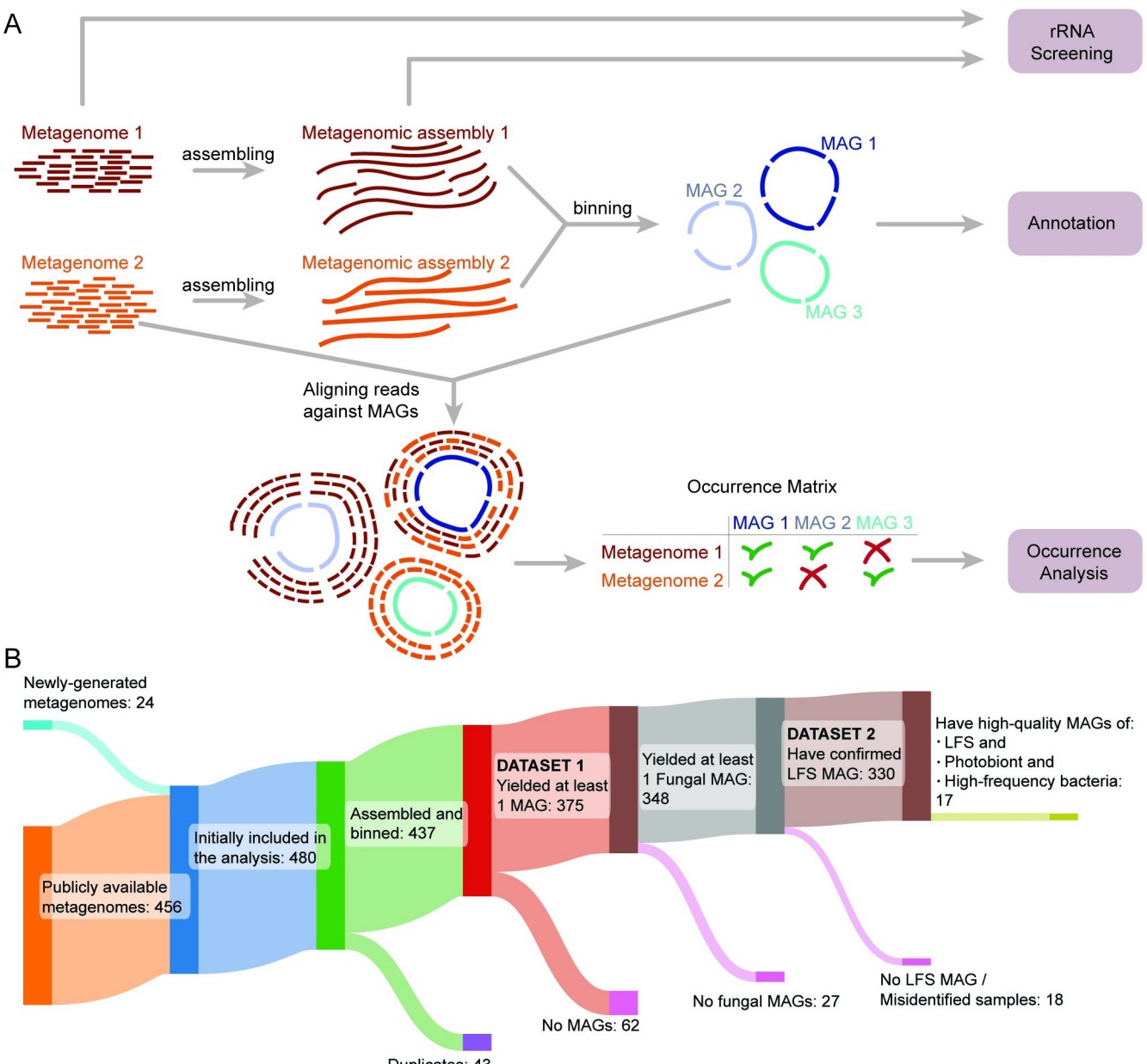

**Fig 1. Bioinformatic pipeline used in the study.** (A) Flowchart of the bioinformatic analysis of this study. (B) Sankey plot showing the source of 437 metagenomes used in the analysis and the progressive reduction of the dataset in the course of the study. LFS, lichen fungal symbiont; MAG, metagenome-assembled genome.

S5 Table). Bacterial taxonomy was assigned according to the Genome Taxonomy Database (GTDB [41]). Fungal MAGs came from 7 taxonomic classes (Fig 2A), with the large majority from the class Lecanoromycetes ($n = 260$), followed by Eurotiomycetes ($n = 15$), Dothideomycetes ($n = 5$), Arthoniomycetes ($n = 6$), Lichinomycetes ($n = 4$), Tremellomycetes ($n = 3$), and Cystobasidiomycetes ($n = 1$) and were assigned in a curation process following [42]. In addition to LFSs, we recovered 22 non-LFS fungal MAGs, including 4 from the basidiomycete classes Cystobasidiomycetes and Tremellomycetes that have been postulated to be involved in some lichen symbioses (S4 Table). Three clades were recovered of Trebouxiophyceae algae,

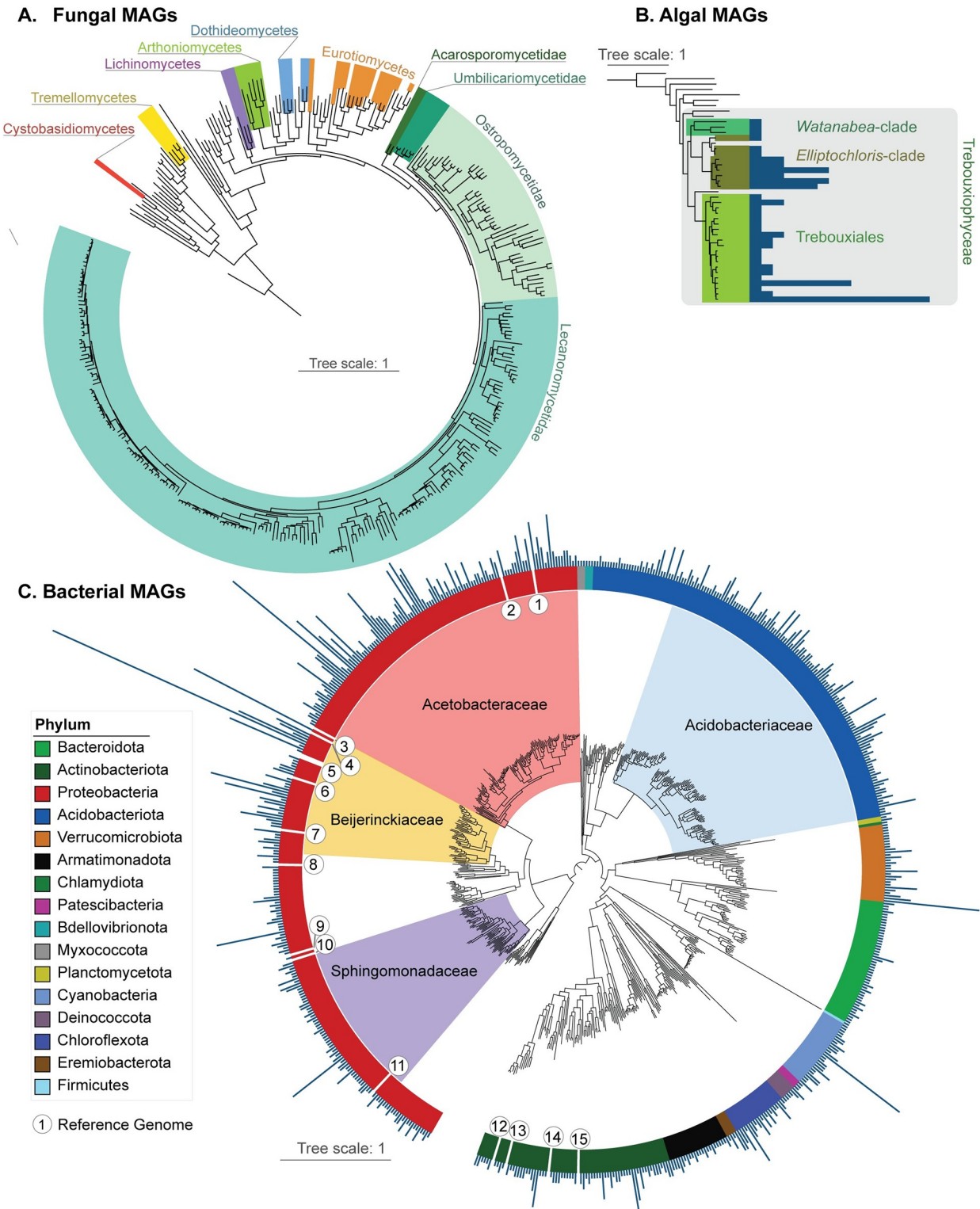

**Fig 2. Maximum likelihood phylogenetic trees of the MAGs.** The trees are calculated using IQ-TREE and are based on alignments of marker genes. Clade labels and color bars indicate broad groups the MAGs were assigned to. Reference genomes (interleaved) are not colored or labeled. (A) Tree of recovered fungal MAGs (colored sectors) interleaved with reference genomes (listed in S5 Table), based on 709 BUSCO genes. (B) Tree showing recovered algal MAGs interleaved with reference genomes based on 1296 BUSCO genes. Bars represent the number of occurrences of the MAG in Dataset 1. (C) Tree of the bacterial MAGs obtained from lichen metagenomes interleaved with reference genomes based on 120

marker genes from GTDB-Tk. The 4 most frequent bacterial families are denoted by colored sectors. Radiating bars represent the number of occurrences for each given MAG in Dataset 1. Reference genomes belong to bacteria previously isolated from lichens and are shown as numbers: 1: *Lichenicola cladoniae* PAMC 26569; 2: *Lichenicoccus roseus* KEBCLARHB70R; 3: *Lichenihabitans minor* RmlP026; 4: *Lichenihabitans ramalinae* RmlP001; 5: *Lichenihabitans psoromatis* PAMC 29128 and PAMC 29148; 6: *Lichenifustis flavocetrariae* BP6-180914; 7: *Methylobacterium planeticum* YIM 132548; 8: *Aureimonas leprariae* YIM 132180; 9: *Rubellimicrobium rubrum* YIM 131921; 10: *Paracoccus lichenicola* YIM 132242; 11: *Polymorphobacter megasporae* PAMC 29362; 12: *Subtercola lobariae* CGMCC 1.12976; 13: *Luteimicrobium album* NBRC 106348; 14: *Streptomyces lichenis* LCR6-01; 15: *Nakamurella leprariae* YIM 132084. Full versions of the phylogenomic trees are available in FigShare (10.6084/m9.figshare.27054937). GTDB, Genome Taxonomy Database; MAG, metagenome-assembled genome.

which include major lichen photobionts (Fig 2B). Bacteria came from 16 phyla, with 42% of species-level lineages ($n = 282$) coming from Proteobacteria and 23% ($n = 153$) coming from Acidobacteriota. Cyanobacteria (a photobiont group) were represented by 27 species. The rest came from: Actinobacteriota ($n = 69$), Bacteroidota ($n = 47$), Verrucomicrobiota ($n = 29$), Armatimonadota ($n = 23$), Chloroflexota ($n = 21$), Deinococcota ($n = 6$), Eremiobacterota ($n = 4$), Bdellovibrionota ($n = 3$), Myxococcota ($n = 3$), Patescibacteria ($n = 3$), Planctomycetota ($n = 2$), Chlamydiota ($n = 1$), and Firmicutes ($n = 1$) (Fig 2C).

## Mapping MAG occurrences reveals a few bacterial lineages to be disproportionately frequent in metagenomes

Given the known propensity of lichens with different LFSs to share, e.g., the same photobiont species, we expected that the same species could occur in multiple metagenomes. We accordingly mapped occurrences of the 95% ANI "species" by read mapping each metagenome against the dereplicated MAGs, where we counted a species as present if over 50% of the MAG was covered (hereafter referred to as "MAG recovery"). MAG mapping resulted in a stepwise reduction of the overall data set of metagenomes used for analysis (Fig 1B). In 62 of the metagenomes, no MAG could be recovered that passed the mapping thresholds outlined above, leaving 375 metagenomes with at least 1 recoverable MAG (Dataset 1). Of these, 348 contained at least 1 fungal MAG. The placement of these fungal MAGs in phylogenomic trees allowed us to identify 18 metagenomes where the taxonomy of the fungal MAG did not match the name of the lichen given in the NCBI metadata associated with the sample, indicating that the sample was likely misidentified (S6 Table). The MAGs from these 18 metagenomes were excluded from the final tree (Fig 2A) and occurrence heatmaps (see below), but included in the overall MAG counts. This left 330 metagenomes that contained a verified MAG of the LFS (Dataset 2). Echoing the taxonomic breakdown of fungal MAGs, the 330 metagenomes represented in Dataset 2 were made from lichen symbioses involving LFSs from 5 putative origins of lichen symbiosis within Ascomycota, including Arthoniomycetes ($n = 6$), Dothideomycetes ($n = 3$), Lichinomycetes ($n = 3$), Eurotiomycetes ($n = 6$), and Lecanoromycetes ($n = 312$).

The mapping of MAG occurrences also allowed us to reconstruct the frequency of individual MAGs across Dataset 1. As expected, most fungal MAGs occurred only once, except in a few cases where several metagenomes of the same lichen symbiosis, or several symbioses with closely related LFSs, were included. For algae and bacteria, we plotted occurrence frequency for each MAG as a column on its respective tip in the phylogenomic tree (Fig 2B and 2C). In algae, the most frequently occurring MAG, from the genus *Trebouxia*, was found 16 times (S4 Table). Bacterial MAG occurrences were highly skewed, with more than 50% of bacterial occurrences deriving from 2 phyla (Protobacteria and Acidobacteriota) and specifically 4 bacterial families, Acetobacteraceae, Beijerinckiaceae, Sphingomonadaceae, and Acidobacteriaceae. Major occurrence spikes can be seen in the genus *Lichenihabitans* in the Beijerinckiaceae (occurred in 99 metagenomes, with 1 species detected in 52 metagenomes); the clade provisionally assigned in GTDB to the genus-level lineage LMUY01 in the Acetobacteraceae

(occurred in 50 metagenomes, with 1 species detected in 28 metagenomes); and CAHJXG01 (Acetobacteraceae), which occurred as various species in 58 metagenomes (S7 Table). High occurrence-count genera outside these 4 families included *Nostoc* in the Cyanobacteria and CAHJWO01 in Verrucomicrobiota (S7 Table). Each high frequency bacterial lineage derived from multiple of the 12 studies from which metagenomes were used, and at minimum was present in data from 3 independent studies produced by different investigators (S8 Table; see also S3 Fig).

To explore how lineages co-occur within lichen samples, we built co-occurrence network graphs (S4 Fig), using the occurrence matrix. We defined co-occurrence as an instance of 2 lineages occurring together in one metagenome. For this analysis, we focused on the groups that are known to stably occur in lichens (algae, Cyanobacteria, and Cystobasidiomycetes and Tremellomycetes fungi), and on the most frequent bacterial groups (most frequent genera of Acetobacteraceae, Beijerinckiaceae, and Acidobacteriaceae). Only metagenomes that yielded an LFS MAG (*n* = 330; Dataset 2) were included in this analysis.

## Known lichen photobionts are underrepresented as MAGs, but detectable in metagenomic raw reads and metagenomic assemblies

In the next analysis, we compared MAG occurrences against a priori known photobiont occurrence patterns and major clades of "host" LFSs. Specifically, we plotted frequencies for the major photobiont lineages Trebouxiophyceae, Ulvophyceae, and Cyanobacteria as proxies for the 3 most commonly assigned photobiont categories in lichen research ("trebouxioid," "trentepohlioid," and "cyanobacteria"). In addition, we plotted data for 2 basidiomycete fungal groups recovered as MAGs in the initial binning exercise and widely discussed as putative lichen symbionts (Cystobasidiomycetes and Tremellomycetes), and the 4 highest frequency bacterial families based on MAG mapping in the previous section. The purpose of this analysis was 2-fold: first, it would allow us to assess the frequency at which MAGs were recovered from known symbionts—photobionts—visible by microscopy; and second, it could reveal if any microbial lineages exhibit co-occurrence patterns with certain photobiont or LFS clades. MAGs of known photobionts are recovered at a low rate compared to their known occurrences (Fig 3). MAGs of known photobionts occurred together with the LFSs with which they were expected to occur, but likewise at low frequency compared to expectations (Fig 3). MAGs of basidiomycete fungi were recovered in only a few metagenomes. The 4 bacterial families that account for most MAGs are recovered at varying frequencies as MAGs across most lichen symbioses, mapped both by LFS and by the major photobiont group (Fig 3A).

We suspected that the low frequency of MAG recovery of photobionts compared to their known occurrences was due to sequencing depth being insufficient to yield photobiont MAGs. To address this limitation, we screened raw read sets of each metagenome for small subunit (SSU) rRNA sequences for all of the same organismal groups for which we had plotted MAG frequency. We also examined SSU rRNA detection at the level of assembly, as we considered that this could represent a conservative estimate. SSU rRNA detection finds the 3 main known photobiont groups Trebouxiophyceae, Ulvophyceae, and Cyanobacteria in most assemblies and read sets (Fig 3A). We detected SSU rRNA genes from major photobiont lineages in all major LFS groups they were expected to occur in, as well as, unexpectedly, in many in which they are not recognized photobionts. As expected, assemblies tended to have lower detection rates than raw read sets, but even with assemblies, photobionts were detected in many lichens they are not considered to occur in. Based on read sets, the Trebouxiophyceae were found in almost all lichen metagenomes (>99%), including in lichens considered to have only a cyanobacterial or only ulvophycean photobiont. Similarly, Cyanobacteria were detected in about

A  Canonical symbionts and major additional organismal group

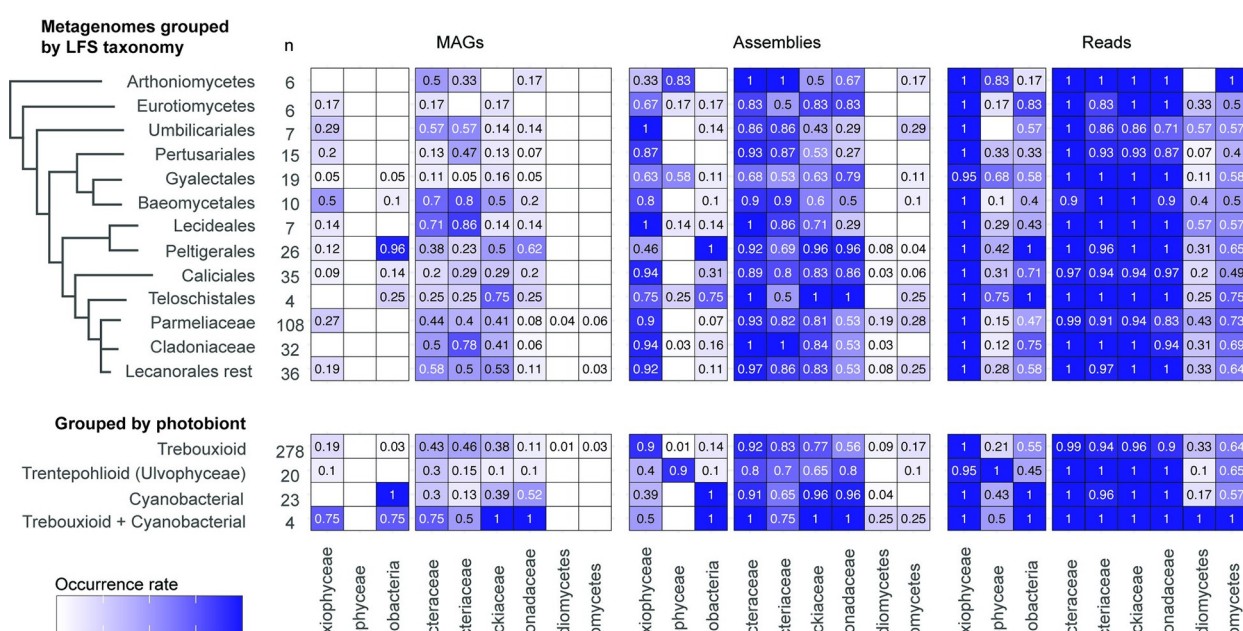

B  Major bacterial genus-level groups

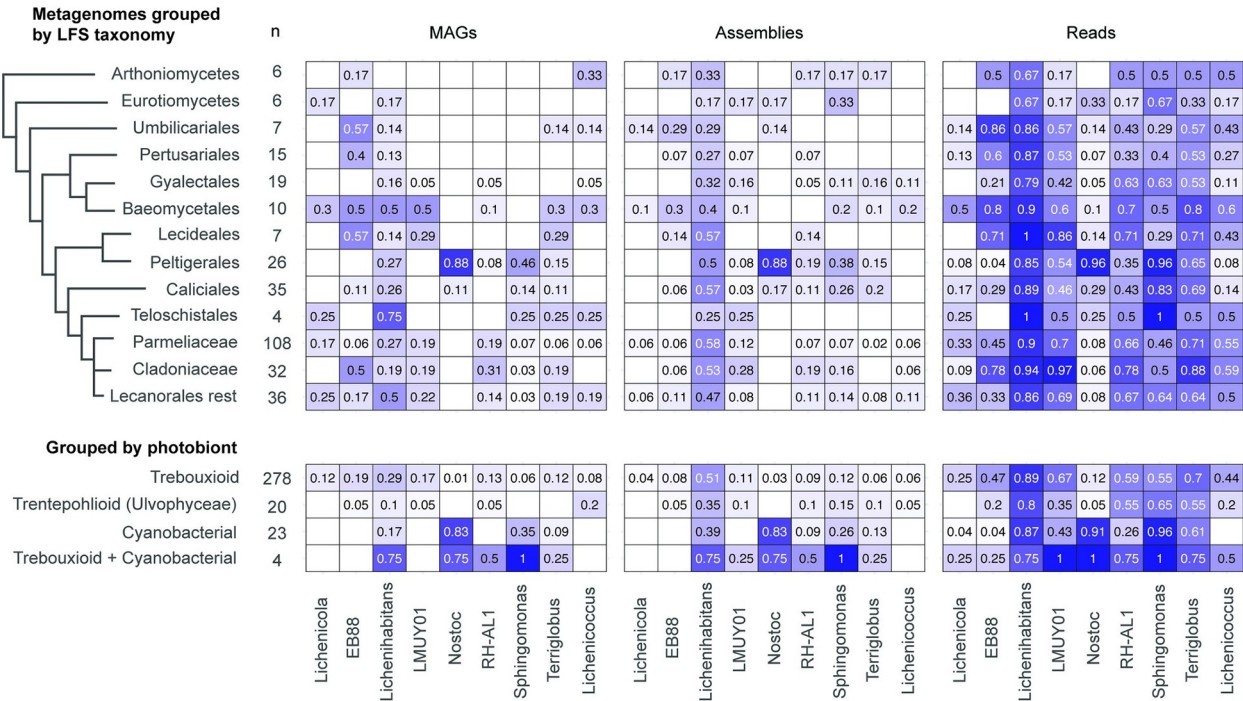

**Fig 3. Organismal occurrence by major microbial group.** Heatmaps showing occurrence rates of mapped MAGs, and detection of canonical symbionts and major additional organismal groups using rRNA in assemblies and raw read sets. (A) For class-level taxa against major LFS (lichen fungal symbiont) groups and major photobiont groups; (B) for 9 of the 13 key genus-level bacterial groups against major LFS groups and major photobiont groups. We grouped metagenomes by the photobiont that is expected in a given lichen based on literature sources (see Supporting information for Spribille and colleagues [9]). Only metagenomes from the Dataset 2 (*n* = 330) are included in this analysis, and only groups

represented by 4 or more metagenomes are shown. The number of metagenomes in each category is shown in the "n" column. The data underlying this figure can be found in S1 Data. LFS, lichen fungal symbiont; MAG, metagenome-assembled genome.

half of metagenomes of lichens considered to have only Trebouxiophyceae or only Ulvophyceae as photobiont ($n$ = 154 out of 278 and 9 out of 20, respectively, based on Dataset 2), and Ulvophyceae were detected in 21% ($n$ = 59 out of 278) and 43% ($n$ = 10 out of 23) of metagenomes of lichens considered to have only Trebouxiophyceae or Cyanobacteria as the sole photobiont, respectively. Similarly, the fungal groups Cystobasidiomycetes and Tremellomycetes occurred at much higher frequencies in assemblies (8% and 15%) and read sets (31% and 64%), respectively, than as MAGs. Although some trends are visible, we did not perform statistics on relative occurrence frequencies owing to uneven sequencing depth across the data set (see Discussion), but some patterns of presence correspond to expectations, such as the high frequency of Cyanobacteria associated with the LFS order Peltigerales.

Bacteria also exhibit a higher detection rate as SSU rRNA than as MAGs. At the level of the detection of SSU rRNA, Acetobacteraceae were detected in assemblies and read sets of nearly all metagenomes, followed in frequency by Beijerinckiaceae, Acidobacteriaceae, and Sphingomonadaceae (Fig 3A). Few patterns relative to major LFS clade or photobiont group are visible, with the exception of high frequency occurrences of Sphingomonadaceae associated with Cyanobacteria and several LFS orders both with (Peltigerales) and without (e.g., Caliciales) recognized cyanobacterial photobionts.

To probe another level of taxonomic resolution, we also mapped 9 of the top 13 bacterial genera from which we selected representatives for annotation (see below; Fig 3B). Four GTDB-classified "genera" (CAHJXG01, Acetobacteraceae gen. sp., CAHJWO01 and CAHJWL01) could not be screened using SSU rRNA as they are currently only represented by MAGs that lack rRNA annotations. As expected, *Nostoc* is recovered at a high occurrence rate in LFS groups known to associate with it as a photobiont, as well as in cyanobacterial symbioses; the few cases of cyanobacterial symbioses where *Nostoc* was not detected involve cases where LFSs associate with other cyanobacterial genera, such as *Scytonema* or *Stigonema*. Of the mapped genera, *Lichenihabitans* is the most frequently recovered MAGs and detected in assemblies and read sets as SSU rRNA. More detailed taxonomic co-occurrence tables are provided in S7 Table.

The publicly available metagenomes used for this survey span a wide range of sequencing depth, measured as sequenced base pairs (S1 Table). Plotting MAG recovery against sequencing depth (Fig 4A and S9 Table) shows a positive relationship, suggesting that no inferences about true absence can be drawn from the numerous shallowly sequenced metagenomes.

When detection of MAGs and SSU rRNA of the 7 mapped groups used in Fig 3 is plotted against sequencing depth, it becomes evident that some detections happen frequently at low sequencing depth, while in contrast, in numerous cases lack of detection persists despite high sequencing depth. The highest rate of MAG recovery was for the bacterial families Acetobacteraceae, Acidobacteriaceae, and Beijerinckiaceae (Fig 4B). For these 3 families, we recovered MAGs in many metagenomes even with low sequencing depth. Far fewer MAGs were recovered at low sequencing depth for the eukaryotes (Trebouxiophyceae or either of the basidiomycete groups). Overall, Trebouxiophyceae and Acetobacteraceae were the most frequently detected as SSU rRNA in raw reads, each having 99% ($n$ = 374 and 372 respectively; based on the Dataset 1) occurrence frequency (Fig 4B), followed by Beijerinckiaceae and Acidobacteriaceae with 97% and 95% (S10 Table). The only cases where SSU rRNA was not detected for Trebouxiophyceae and Acetobacteraceae was in metagenomes with low sequencing depth, suggesting that the occurrence of these 2 taxon groups cannot be ruled out in those

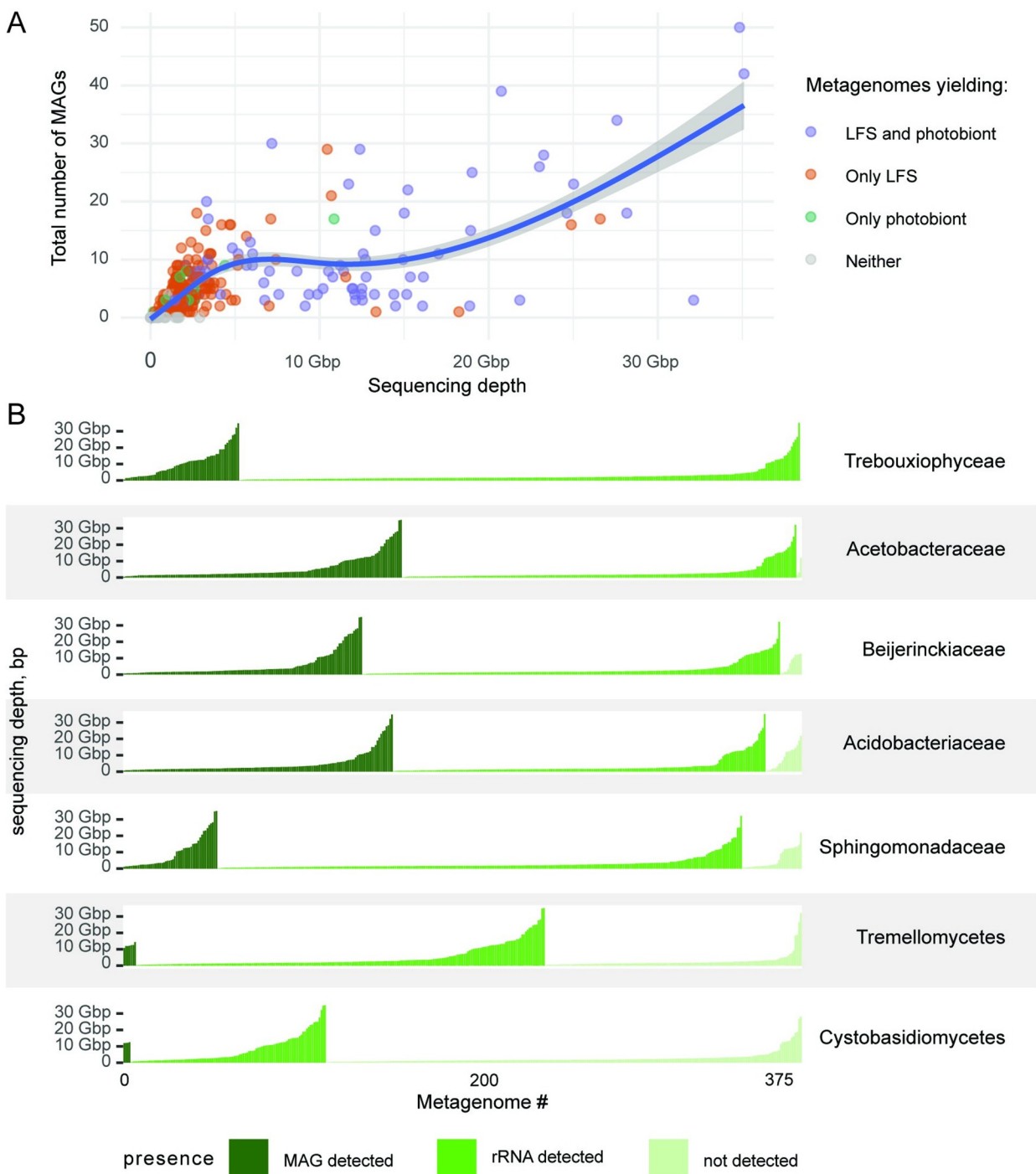

**Fig 4. Organism detection as a function of sequencing depth.** (A) Number of recovered MAGs as a function of sequencing depth (bp). Each dot represents a metagenome colored based on the recovery of the LFS (lichen fungal symbiont) and the photobiont. The curve indicates a generalized additive model (GAM) smoothing. The data underlying this figure can be found in S9 Table. (B) Detection of the key microbial groups in lichen metagenomes based on 2 methods of screening: presence of MAGs and presence of the SSU rRNA in the raw, unassembled reads as a function of sequencing depth. Each vertical bar represents a single metagenome, with the bar heights representing sequencing depth and the color representing the screening outcome. The metagenomes are sorted along the x-axis based on the outcome of the screening and the sequencing depth. The data underlying this figure can be found in S1 Data. GAM, generalized additive model; LFS, lichen fungal symbiont; MAG, metagenome-assembled genome; SSU, small subunit.

metagenomes. The 2 basidiomycete classes Tremellomycetes and Cystobasidiomycetes were detected in about two thirds and one third of read sets, respectively ($n$ = 234 and 112, respectively; Fig 4B). For these, we failed to detect SSU rRNA in numerous deeply sequenced metagenomes (>10 Gbp; mapped as "not detected" in Fig 4B) suggesting that these lichens genuinely lack these taxa.

## Recently published data validate high frequency of 4 bacterial families

In light of new lichen metagenomes being constantly deposited by researchers, we validated the occurrence patterns of high-frequency bacterial families in 243 additional metagenomes not included in the initial data set (S11 Table). These derived from 4 taxonomic groups of fungal "hosts": Lichinomycetes ($n$ = 17, as circumscribed by Díaz-Escandón and colleagues [43]), Dothideomycetes ($n$ = 3), Arthoniomycetes ($n$ = 2), and Lecanoromycetes ($n$ = 221). Of the Lecanoromycetes, 157 were derived from just 3 LFS genera owing to inclusion of studies that included replicated sampling of those lichens (*Cladonia*, $n$ = 51; *Rhizoplaca*, $n$ = 73; and $n$ = 33 from the species *Lecanora polytropa*; S11 Table) and 62 came from replicates of lichens involving just 2 species of LFS. Of the newly generated metagenomes, 95% (231 of 243) come from LFS families already included in the initial data set. Screening raw unassembled reads for SSU rRNA sequences for these data and tabulating occurrence frequencies of major taxa yielded results consistent with our data set (S12 Table): the same 4 bacterial families have the highest prevalence, and the prevalence of Acetobacteraceae reaches 97.1%. *Lichenihabitans* was detected in 67.5% of these metagenomes.

## None of the non-cyanobacterial bacteria are predicted to fix nitrogen and most are carbon heterotrophs

The availability of numerous new MAGs offered the possibility to compare predicted metabolic capabilities of the highest frequency bacteria in and on lichens, irrespective of whether they are ultimately found to be constitutive components of the symbiosis. We rank-ordered bacterial genera by the number of occurrences and selected the 13 most frequent genera, which together accounted for 37% of bacterial lineages and 53% of bacterial occurrences, occurring in a total of 252 metagenomes. Next, we annotated all MAGs assigned to these genera that were near complete genomes (≥95% complete and ≤10% contaminated) (Fig 5A and S13 Table), for a total of 63 out of 250 MAGs (detected in 122 metagenomes). We focused on 5 selected metabolic capabilities that have been addressed in previous work on bacteria from lichens: (1) nitrogen fixation; (2) carbohydrate modification and transport, as well as modification of exogenous carbon sources; (3) anoxygenic photosynthesis; (4) iron scavenging and provisioning; and (5) cofactor synthesis.

The involvement of non-cyanobacterial bacteria in nitrogen fixation is one of the oldest hypotheses in lichen microbiology. Hyphomicrobiales (Rhizobiales) bacteria in lichens have been postulated to be involved in nitrogen fixation since the 1930s [44], and in recent decades several authors (Liba and colleagues [45], Grube and colleagues [11], Hodkinson and Lutzoni [46], Almendras and colleagues [47]) reported detecting *nifH* from lichen bacteria. However, we did not recover *nifH*, the key gene required for fixing nitrogen, in any non-cyanobacterial MAGs. To account for the possibility that *nifH* is present on a plasmid and therefore failed to be detected in MAGs passing threshold, we searched for *nifH* across all metagenomic assemblies using multiple queries (S14 Table). Furthermore, the inability to detect *nifH* was not an artifact, as we recovered it using tblastn with high frequency in Hyphomicrobiales genomes that we surveyed from Genbank, and that we used to construct a phylogenomic tree to contextualize the placement of lichen-derived Hyphomicrobiales (S5 Fig). We found only 1 non-

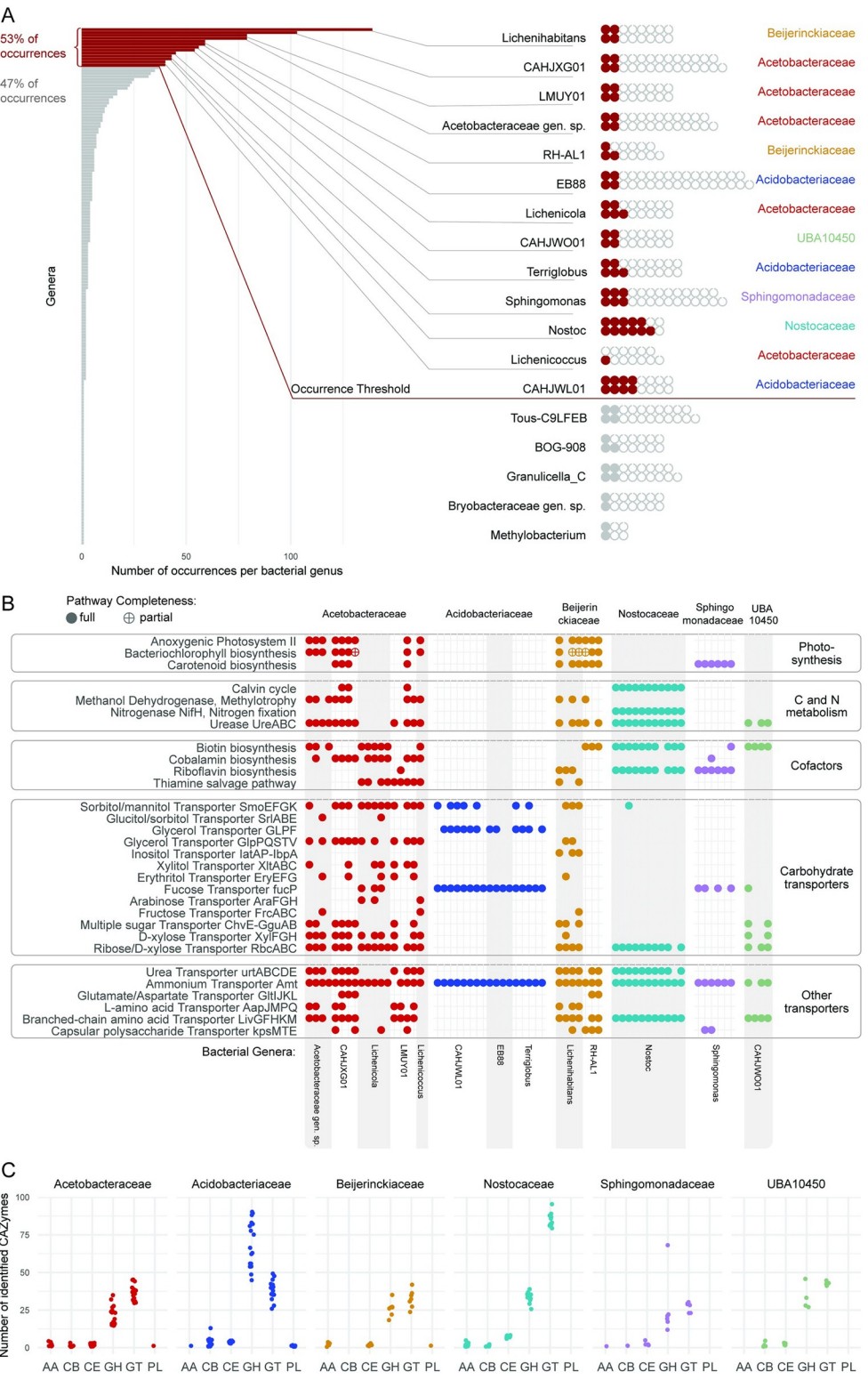

**Fig 5. Functional annotation of key bacterial MAGs.** (A) Selecting MAGs for annotation. The bar graph on the left shows the total number of occurrences per bacterial "genus" (after GTDB), with genera listed in decreasing order. Red bars represent the genera selected for functional annotation, which were the 13 most frequent genera together accounting for 53% of all bacterial occurrences. The waffle graph on the right shows the most frequent bacterial genera and MAGs assigned to them. Red circles represent MAGs selected for annotation: all MAGs from the selected genera

that had >95% completeness. Filled gray circles represent MAGs with >95% completeness that belonged to less frequent genera and were not annotated. The open gray circles represent MAGs with ≤95% completeness, also not annotated. (B) Presence of selected pathways and protein complexes in the MAGs of most common lichen bacteria. Each column represents one of the 63 annotated MAGs, grouped by their taxonomy. We reconstructed pathways using KEGG; to detect carotenoid BGCs, we used antiSMASH. Here, we show the presence of pathways and protein complexes potentially relevant to the symbiosis. For 3 pathways (biosynthesis of bacteriochlorophyll, biotin, and cobalamin), we also show partial completeness (allowing one missing gene). (C) Number of genes assigned to each CAZy class per MAG. We annotated CAZymes in the MAGs selected for an in-depth annotation using dbcan. The data here are grouped on the family level. The CAZy classes are: Auxiliary Activities (AA), Carbohydrate-Binding Modules (CB), Carbohydrate Esterases (CE), Glycoside Hydrolases (GH), Glycosyl Transferases (GT), and Polysaccharide Lyases (PL). The data underlying this figure can be found in S1 Data. GTDB, Genome Taxonomy Database; MAG, metagenome-assembled genome.

cyanobacterial *nifH*, from Hyphomicrobiales, in one metagenome. By contrast, nearly all analyzed lineages possess the ammonium transporter *amtB* (Fig 5B). In addition, in 3 families (Acetobacteraceae, Beijerinckiaceae, and Nostocaceae), the majority of studied MAGs had genes of the urea transporter *urtABCDE* and urea metabolism *ureABC* (Fig 5B). Most lineages possess various amino acids transporters: *uivGFHKM* for branched-chain amino acids, *aapJMPQ* for L-amino acids, and *GltIJKL* for glutamate/aspartate (Fig 5B).

In addition, we examined iron scavenging, briefly suggested by [48] to be a role played by lichen bacteria, and a survival function of both host-associated and unassociated bacterial communities [49,50]. Gene clusters potentially involved in siderophore biosynthesis were rare in all bacterial groups except Cyanobacteria. Even though every analyzed MAG had genes related to siderophore transport (*tonB*-dependent receptors), these genes are not exclusively connected to, and cannot be viewed as evidence of, siderophore uptake [51]. Instead, the majority of annotated MAGs (78%; *n* = 49) encoded iron ion transporters (S15 Table), suggesting that dissolved iron is present and not a limiting nutrient in the system.

Lichen-associated bacteria have been reported to be involved in degradation of eukaryote-derived chitin and glucans [11] as well as hemicellulose and starches [52]. We found gene predictions consistent with these observations as well as suggesting that carbohydrate degradation capacities are unequally distributed among the major bacterial groups. The family Acidobacteriaceae had a markedly different set of predicted carbohydrate-active enzymes (CAZymes) compared to the other bacterial families included in the annotations. Acidobacteriaceae had twice as many glycoside hydrolases (GH) as the other bacterial families (average = 69 in Acidobacteriaceae, 22–34 in other families). Moreover, only in Acidobacteriaceae were GHs the dominant class of CAZymes (Fig 5C and S16 Table). Acidobacteriaceae possess several GH families targeting mannans, one of the dominant cell wall polysaccharides of LFSs: GH92 (mannosidase), GH125 (exo-alpha-1,6-mannosidase), GH38 (alpha-mannosidase), and GH76 (alpha-1,6-mannosidase/alpha-glucosidase). Also present at a higher frequency in Acidobacteriaceae were GH18 (chitinase), GH55 (beta-1,3-glucanase), and GH51 (endoglucanase, endoxylanase, cellobiohydrolase; S17 Table), all known to act on fungal polysaccharides.

To identify potential carbon sources for bacterial heterotrophs, we analyzed the presence of transporter genes. Acetobacteraceae and one genus of Beijerinckiaceae (*Lichenihabitans*) have a large arsenal of transporters for various monosaccharides (ribose/D-xylose transporter *rbcABC*, D-xylose *xylFGH*, multiple sugars *chvE-gguAB*, more rarely fructose *frcABC* and L-arabinose *araFGH*) and polyols (glycerol *glpPQSTV*, sorbitol/mannitol *smoEFGK*, xylitol *xltABC*, erythritol *eryEFG*, glucitol/sorbitol *srlABE*) (Fig 5B). Other studied bacterial families had fewer transporter systems, although Acidobacteriaceae had a glycerol transporter *GLPF*. Surprisingly, the second genus of Beijerinckiaceae, RH-AL1, did not possess any predicted sugar or polyol transporters, suggesting that these bacteria might not take up sugars. However,

we cannot rule out sugar uptake, since RH-AL1 MAGs encoded several putative transporters with unknown function.

Eymann and colleagues [53] postulated, based on metaproteomic and taxonomic assignment data, that Hyphomicrobiales may be responsible for the catabolism of methanol or other C1 molecules possibly produced as a byproduct of secondary phenol synthesis by the LFS. Several lineages of Acetobacteraceae and Beijerinckiaceae exhibited annotations consistent with methylotrophy, the ability to use methanol as a carbon source. Methanol dehydrogenases (*xoxF* or *mxaF*) were detected in both Beijerinckiaceae genera with annotations (*Lichenihabitans* and RH-AL1), and 5 of 6 Acetobacteraceae genera (Fig 5B), which were present in at least 50 metagenomes. None of the studied bacteria appeared to use methane as none possess methane monooxygenase genes.

Among all annotated MAGs, 29 encoded biosynthetic gene clusters with similarity to known exopolysaccharide-producing clusters (S18 Table; based on annotations with Jaccard distance <0.7 estimated by SanntiS). Lineages with these clusters were detected in 61 metagenomes. In addition, MAGs of several lineages from Acetobacteraceae, Beijerinckiaceae, and Sphingomonadaceae encode a capsule polysaccharide transport system (Fig 5B).

## Photosynthesis is widespread in some bacterial lineages

Pankratov and colleagues [54] recently reported bacteriochlorophyll a from the bacterial genus *Lichenococcus* (Acetobacteraceae) from a *Cladonia* lichen, suggesting a capacity for photosynthesis in non-cyanobacterial lichen bacteria. Fifteen of our annotated bacterial MAGs possessed both a complete set of anoxygenic photosystem II proteins (KEGG modules M00597 and M00165; *pufABCML-puhA*), and a bacteriochlorophyll biosynthesis pathway (*acsF*, *chlBNL*, *bchCFGPXYZ*). Many also contained carotenoid biosynthetic gene clusters (Fig 5B). All of these MAGs came from Acetobacteraceae and Beijerinckiaceae (including *Lichenihabitans*). This combination of pathways corresponds to a bacterial group known as aerobic anoxygenic phototrophs (AAPs [55]). The AAP profile is present in 6 of the 7 annotated Beijerinckiaceae MAGs, and in 9 of the 18 annotated Acetobacteraceae MAGs. The MAGs that possessed these predicted features were mapped as occurring in 56 of the 122 metagenomes for which any annotated bacterial MAG was available (15% of metagenomes from Dataset 1).

Pankatrov and colleagues also characterized lichen bacteria with both a partial [54] and complete [56] Calvin–Benson(-Bassham; CBB) pathway from lichen-derived Acetobacteraceae and Beijerinckiaceae, respectively, suggesting that at least some photosynthetic bacteria also fix carbon. Fourteen MAGs possessed a complete CBB cycle (Fig 5B and S1 File). These included all 11 cyanobacterial photobionts, which are known autotrophs, and 3 species in Acetobacteraceae. None of the remaining bacterial MAGs (*n* = 49) were predicted to encode a complete alternative carbon fixation pathway [57]: reductive citrate cycle (KEGG module M00173), 3-hydroxypropionate bi-cycle (M00376), hydroxypropionate-hydroxybutyrate cycle (M00375), dicarboxylate-hydroxybutyrate cycle (M00374), Wood–Ljungdahl pathway (M00377), or the phosphate acetyltransferase-acetate kinase pathway (M00579), and all are therefore likely to be carbon heterotrophs.

## Most high frequency bacterial lineages are predicted prototrophs of some B vitamins

Bacteria have been postulated to synthesize essential cofactors required by algal symbionts [58], and bacterial-derived cofactors have been detected in a metaproteomic study of the lichen *Lobaria pulmonaria* [58,59]. A complete biotin synthesis pathway (complete module M00123 or the alternative M00950) were present in 43% of the analyzed bacterial MAGs (*n* = 27; Fig 5B

and S1 File). None of the MAGs contained a complete thiamine biosynthesis pathway, but the thiamine salvage pathway was complete in several Acetobacteraceae and Beijerinckiaceae (KEGG Module M00899 or *thiMDE* [60]). Several Acetobacteraceae and 1 Sphingomonadaceae MAG encode a pathway for synthesis of cobalamin (vitamin B12; KEGG Module M00122). Taken together, the MAGs that possessed features of some form of B vitamin prototrophy were mapped as occurring in 99 of the 122 metagenomes for which any bacterial annotated MAG was available (26% of all metagenomes from Dataset 1).

## B-vitamin auxotrophy is common in lichen fungi

The recovery of widespread biotin and cobalamin prototrophy in annotated bacterial MAGs, and the frequency with which corresponding bacterial lineages were mapped onto metagenomes, prompted us to annotate all high-quality eukaryotic MAGs derived from the present study to screen for potential auxotrophies. For this, we selected all nearly complete eukaryotic MAGs (≥95% complete and ≤10% contaminated), resulting in 9 algal MAGs, 71 LFS MAGs (including 3 LFS MAGs from misidentified samples) and 4 MAGs of non-LFS fungi (S2 File).

All 9 annotated lichen algal MAGs are predicted to possess at least partially complete pathways for thiamine synthesis and scavenging and biotin synthesis. Of them, 4 are predicted to have a complete pathway for thiamine synthesis and 7 to have a complete thiamine scavenging pathway (Fig 6). By contrast, 8 of the 9 algal MAGs lack most of the predicted components of cobalamin metabolism and one lacks all genes (Fig 6). The missing components correspond to 10 out of 11 enzymes in the module M00925 and 6 out 7 enzymes in the module M00122 (S2 File). To establish whether lichen algae actually need cobalamin, we screened their MAGs for the cobalamin-dependent methionine synthase gene *metH*, as well as its cobalamin-independent alternative *metE*. While five of the algal MAGs possessed only *metE*, the other 4 possessed both *metH* and *metE* (S19 Table). This result echoes that from Croft and colleagues [61], who suggested that some algae preferentially use more efficient *metH* in the presence of cobalamin, and switch to *metE* only when cobalamin is absent.

Fungal MAGs exhibit a mixed pattern of predicted thiamine synthesis and/or scavenging systems. Only 2 MAGs are predicted to have a complete thiamine biosynthesis pathway: an LFS from Arthoniomycetes and an early diverging lecanoromycete lineage in the Acarosporales (Fig 6). Thiamine scavenging pathway was at least partially complete in nearly all fungal MAGs, and a complete pathway was present in one third of MAGs (*n* = 25), which were scattered across the whole fungal tree (Fig 6). Notably, all 8 annotated members of the fungal order Peltigerales lack both thiamine synthesis and salvage pathways, with all components of the former missing except thiamine pyrophosphokinase (S2 File).

All 4 of the non-LFS fungal MAGs plus LFS from Eurotiomycetes, Arthoniomycetes, Dothideomycetes, and the early diverging lecanoromycete lineage Umbilicariales are the only fungal lineages to possess partial predicted biotin pathways. All other fungi, all of which are LFSs, lack any predicted component of biotin synthesis. No components of cobalamin synthesis were detected in the fungal MAGs and no cobalamin-dependent biochemistry is known from this group of fungi [62].

## Fungal vitamin auxotrophs co-occur with algal and bacterial prototrophs

Thiamine and biotin are essential for life and since they do not appear to be synthesized by some (thiamine; Peltigerales) or most (biotin) LFSs, they will need to be offset from exogenous sources. Based on our annotations, candidates for providing these cofactors could include the alga, cyanobacterium or non-cyanobacterial bacteria, but this depends on the combination of MAG co-occurrences in the sampled lichens as represented by metagenomes. In our final analysis, we identified all lichen metagenomes from which at least 1 complete MAG each from the

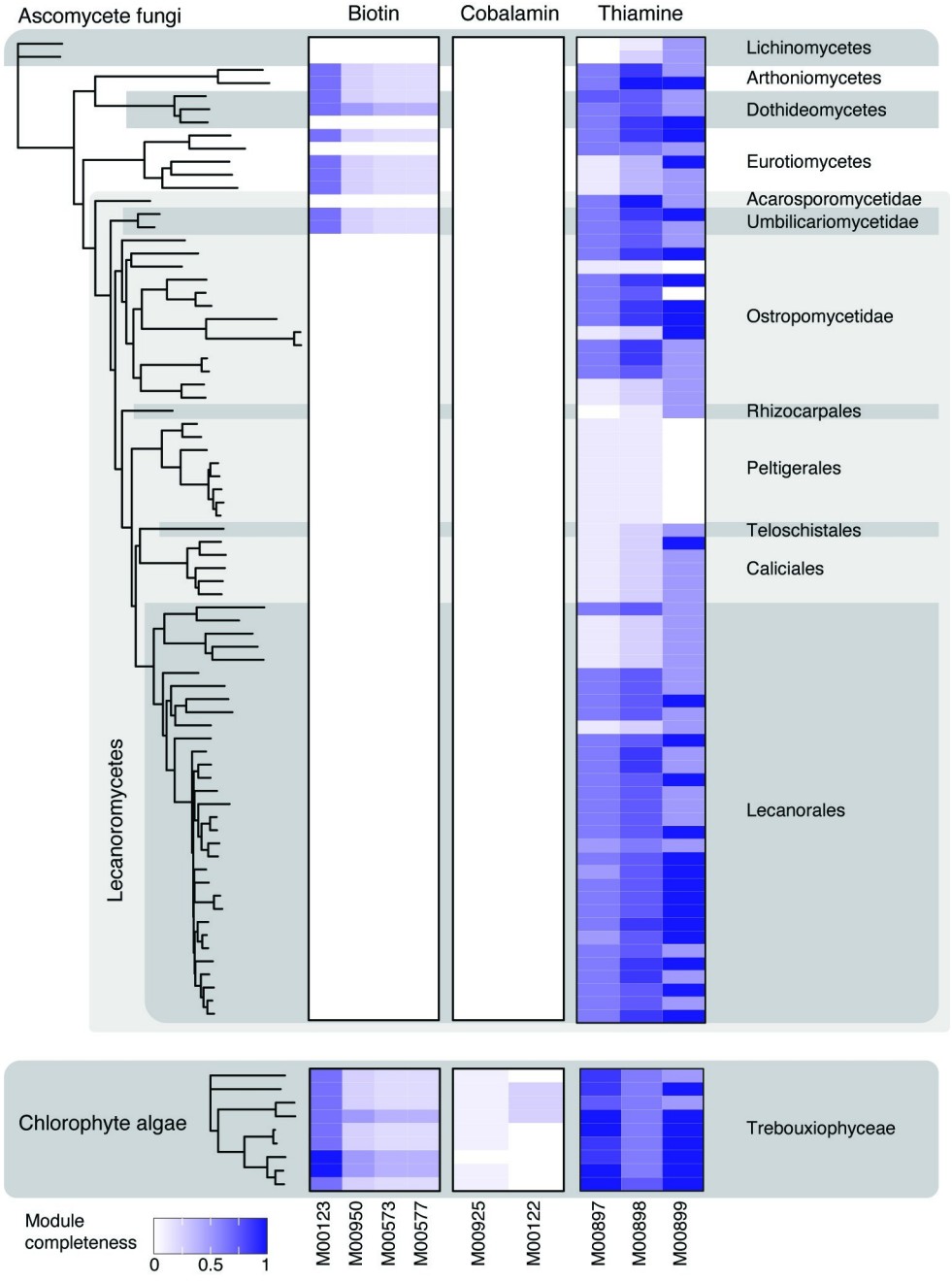

**Fig 6. Completeness of modules involved in cofactor biosynthesis in the high-quality eukaryotic MAGs.** We selected all ≥95% complete and ≤10% contaminated eukaryotic MAGs, and used KEGG to determine completeness of pathways involved in synthesis of biotin, thiamine, and cobalamin. The trees on the left represent the phylogenomic trees constructed for the MAGs; only tips corresponding to the high-quality MAGs are shown. Color on the heatmap represents the percentage of present blocks in a given pathway. The data underlying this figure can be found in S1 Data. MAG, metagenome-assembled genome.

LFS, photobiont and an annotated non-cyanobacterial bacterium was present in ≥95% completeness and ≤10% contamination (from the set of 63, above), to assess in which combinations predicted auxotrophies and prototrophies complemented each other. Only 16 metagenomes fit these criteria (Figs 1B and 7). It should be noted that all of these

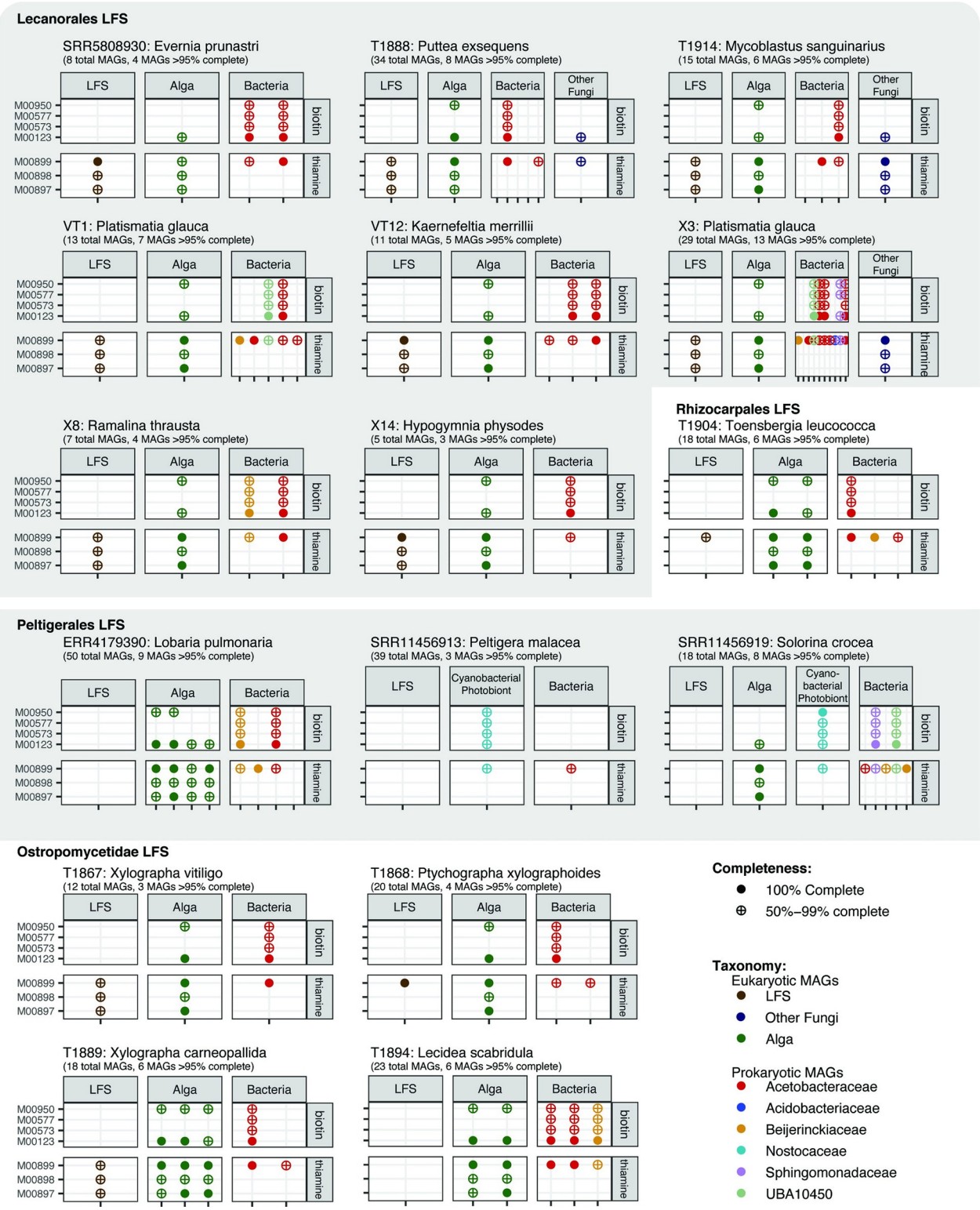

**Fig 7. Co-occurrence of cofactor synthesis modules within lichen metagenomes.** Each panel represents a highly complete lichen metagenome (defined as a metagenome that contains the MAGs of the LFS, photobiont, and at least 1 high-frequency bacterium, all at ≥95% complete and ≤10% contaminated). Each column within a panel represents a MAG, labeled based on the inferred role it plays in the symbiosis and colored based on its taxonomic assignment. For each genome, we show the presence/absence and completeness of several KEGG modules for synthesis of biotin and thiamine. The data underlying this figure can be found in S1 Data. LFS, lichen fungal symbiont; MAG, metagenome-assembled genome.

metagenomes included additional MAGs that did not pass the quality threshold and therefore were not used. The 16 metagenomes include 8 LFSs from the order Lecanorales with *Trebouxia* photobionts, 3 from Peltigerales with a cyanobacterium only ("bipartite") or an alga and cyanobacterium ("tripartite") as photobionts, whereby in *Lobaria pulmonaria* only the alga was recovered as a complete MAG, and 5 additional LFSs from Rhizocarpales and Ostropomycetidae with *Trebouxia* or a member of *Elliptochloris*-clade as photobiont.

The 4 metagenomes with fungal MAGs with thiamine and biotin auxotrophy co-occurred either with an alga with an at least partially complete thiamine and/or biotin pathway and non-cyanobacterial bacterium, or with a cyanobacterium with a partial thiamine synthesis/salvage pathway (Fig 7). An additional 7 metagenomes, in which the LFS MAG contained only a partial thiamine synthesis/salvage pathway, contained a thiamine prototrophic alga and/or bacterium. All 13 metagenomes with a biotin auxotrophic LFS co-occurred with both an algal and a bacterial biotin prototroph.

## Discussion

The reanalysis of lichen metagenomes we present here is a case study in the paradoxes inherent in short read metagenomics data. On the one hand, the information content can be high, and, when MAGs can be assembled, they offer the possibility to extract data that allow the frequency, protein diversity and potential metabolic capacities to be characterized for organisms that have yet to be cultured and in many cases even seen. Our reanalysis of lichen metagenomes resulted in a large and to our knowledge previously unseen body of evidence on organismal presence in lichens: recovery of 1,000 MAGs, two-thirds of which are newly assembled bacterial MAGs; heretofore unavailable documentation of organismal presence in lichen thalli; and annotations and comparative analyses of metabolic potential in 63 bacterial MAGs for which completeness, at ≥95%, and contamination ≤10%, rival genomes derived from culture. On the other hand, current approaches to metagenomics with short reads run up against hard limits when MAGs for various reasons cannot be assembled. As our data also show, even the photobiont, which on account of its size and color can be seen by visual inspection alone, is not recovered as a usable MAG in most lichen metagenomes analyzed. The limitations that arise from assembly and detection of some MAGs place caveats on the ability to assess, with confidence, absence: absence of microbes and absence of predicted proteins. Our study is at its core an attempt to establish (a) what can and cannot be said about microbial occurrence in lichen metagenomes from current short read data, based on what we know is missing; and (b) identify microbes associated at high frequency with lichen symbioses that may represent promising candidates for further research on metabolic interactions.

### Shortfalls in eukaryotic MAG assembly

Organismal occurrence analyses highlight how the combination of community complexity and inadequate sequencing depth can lead to underestimates of even known symbiont occurrence patterns, especially for eukaryotes. In 62 metagenomes—14% of the data set—we failed to recover a single MAG that passed minimum quality thresholds, including of the most abundant organism, the LFS. Even in cases of lichen symbionts that are visible, well-documented and in considerable cellular abundance, as is the case with photobionts, we were able to recover only a few QS50 MAGs, representing only a fraction of all known occurrences. We recovered more cyanobacterial MAGs than those from Trebouxiophyceae, and none from the trentepohlian photobionts of the Ulvophyceae.

The explanations for the assembly of so few QS50 MAGs and their subsequent low MAG recovery rate are at least 2-fold. First, the sequencing depth of the publicly available

metagenomes spans 6 orders of magnitude, with a strong skew to shallowly sequenced metagenomes, which account for nearly all cases in which no LFS MAG was recovered (S6 Fig). Commonly applied sequencing depths and short reads are clearly insufficient to achieve QS50 MAGs for even commonly occurring, undisputed lichen symbionts, and this can be expected to impact assembly regardless of genome characteristics. Second, assembly quality and QC metrics are affected by both genome size and the propensity for closely related genomes to intermingle. The eukaryotic photobionts differ markedly from bacterial photobionts in this regard. The most common Trebouxiophycean genus, *Trebouxia*, with a haploid genome >50 MB in size [27], is well known to occur in individual thalli as a mixture of closely related strains [63,64], which can be expected to introduce graph ambiguities and result in assembly breaks. This may account for why the only MAG that was recovered frequently had a contamination level (i.e., duplicate single copy marker genes) well above the threshold for inclusion in downstream analyses. The most common Ulvophycean genus, *Trentepohlia*, from which no QS50 MAG was obtained, is thought to possess the largest genome of any canonical lichen symbiont, at >100 MB [65,66], is thought to be a polyploid [67,68], and has no published genome at all to date. The cyanobacterium *Nostoc*, by contrast, has smaller genomes than both of these eukaryotic genera, and is not known to occur in multiple strains in lichen thalli (C. Pardo de la Hoz, pers. comm., 2024). We recovered a single *Nostoc* MAG each in all lichens expected to have *Nostoc* as a photobiont in Dataset 2 (Fig 3) and only recovered 2 *Nostoc* MAGs in a single metagenome once, in a *Phaeophyscia* lichen not expected to have *Nostoc* as a photobiont; we assume these grew on the surface of the lichen. These 2 MAGs were phylogenetically divergent with ANI of 90.6%. Whether genome size and occurrence in single strains fully explain the higher MAG recovery rate is unclear. We note that *Nostoc* is also the only microbial component of lichen metagenomes that regularly occurs in higher coverage depth than the LFS (S7 Fig).

Screening assemblies and raw read sets of known SSU rRNA sequences dramatically increased the rate of positive hits from higher order taxonomic groups that correspond to those containing the photobionts, and using raw reads in particular returned hits for all lichen symbioses in which they are already known to occur. However, screening raw reads also returned hits in many symbioses in which they have never been documented using microscopy. For example, positive hits from Trebouxiophyceae were recovered in all but one of the 61 metagenomes from lichens assessed with classical microscopy as having only cyanobacteria or only Ulvophyceae. Cyanobacteria, for their part, were detectable in fully half of lichen symbioses in which only Trebouxiophyceae or only Ulvophyceae are the microscopically documented photobionts. These detections are unlikely to be false positives, as we used a method relying on first extracting all SSU rRNA sequences from the data and then classifying each sequence based on its similarity to the entries in the reference database that includes sequences from a wide range of organisms. To ensure reliability, the classification—and the taxonomic level at which the assignment is provided—were informed by the top 5 hits for each identified SSU rRNA sequence. Instead, we see 2 explanations as most likely to account for these occurrences. First, they could reflect the presence of algal cells in or on the exopolysaccharide mucilages that frequently coat or associate with lichen thalli [69]. The broad taxonomic categories we used (e.g., Trebouxiophyceae) include many free-living aeroterrestrial algae, and the cells in question are not necessarily photobionts in the strict sense. A second possibility is that the detected cells are indeed secondary photobionts of the sampled lichen. Separate methods, including visualizing cell location and cell-to-cell interactions, would be necessary to determine this with certainty, but we note that some secondary algae such as *Coccomyxa viridis* exhibit strong associations with lichens that otherwise contain, e.g., *Trebouxia* photobionts [70].

Similarly, for basidiomycete fungi reported to occur in lichens as yeasts (classes Cystobasidiomycetes and Tremellomycetes), we recovered only a few MAGs. Based on SSU rRNA screening, the detection rate increased to about one-third of metagenomes for Cystobasidiomycetes and two-thirds for Tremellomycetes. Using PCR screening, Cystobasidiomycetes were originally reported from 75% of macrolichens in which a fungus of the Parmeliaceae ("parmelioids") constitutes the LFS [12]. Compared to this, the 41% detection rate from SSU rRNA screening of parmelioids in the present study is low. A possible explanation for this is different methods used—targeted PCR-based screening versus presence of rRNA in metagenomic libraries—and the inclusion of libraries with comparatively shallow sequencing depths in our data set. The detection rate is however much higher than the nine lichens in which they were detected in the 339 metagenomes of Lendemer and colleagues [28], which are part of this study. The low detection rate of basidiomycete yeasts in the latter study is almost certainly related to the shallow sequencing depths of many of the metagenomes used in that study, which included all 62 of the metagenomes we excluded due to recovery of no MAGs and 26 of 27 of the remaining metagenomes from which no LFS MAG could be recovered (see Fig 1B). In addition, low sequencing depth can be expected to affect the ability to capture even rRNA of organisms such as Cystobasidiomycetes occurring in low cellular abundance (in this case, typically ca. 1:100 depth coverage ratio to the LFS: S7 Fig).

## What can and cannot be said about the bacterial fraction

By far the largest fraction of assembled MAGs is contributed by bacteria, with 674 non-redundant species MAGs. These MAGs, and their associated mapping data and annotations, potentially represent a data windfall for the study of lichen-associated bacteria. However, the bacterial composition of the extracted data can be expected to include not only any bacteria that have been postulated to play biologically meaningful roles in healthy lichens, but also any involved in degradation of senescing thalli, by-catch from the surrounding environment and contaminants introduced during sampling. Because the dataset is inherently heterogeneous, the limits of inference can be expected to differ depending on the questions being asked, the intrinsic attributes of the data sets themselves and basic assumptions about lichen systems. Our approach here was to recursively analyze progressively smaller datasets, defined by quality metrics (Fig 1B), and, as in our previous work [12,14], limit our analyses to the most frequently occurring lineages on the assumption that the likelihood that a lineage occurrence is attributable to contamination decreases as its detection frequency increases across independent samples.

At least 3 data set attributes broadly framed the limits of inference for our objectives. First, the collection of the original material was not guided by considerations about bacterial composition. This could be seen on the one hand as an advantage, as the sampling is blind, but on the other hand it also means that sampling sources (e.g., lobe tips or fruiting bodies), which could be expected to differ in bacterial composition, were not specified (except in [14,31,36,40] and the 24 new metagenomes). Second, the data set is taxonomically broad, meaning that specific bacterial lineages that may be important for certain lichen symbioses would occur in only low frequency. Third, some of the same constraints that affect the interpretation of the eukaryotic data, such as sequencing depth, also affect the bacterial set, with both MAG recovery frequency and quality likely to be lower in shallowly sequenced metagenomes. As a result of these 3 built-in constraints, whatever patterns that emerge would likely be a "lowest common denominator" across the broader swathe of sampled lichen symbioses.

We ultimately determined that the structure, sample handling and sequencing protocols underlying the data set would, with appropriate filtering, permit inferences regarding bacterial

presence; sample-level taxonomic composition (as lichens were sampled blindly with respect to their bacterial composition); minimum overall frequency of occurrence (both as MAGs and 16S rRNA); predicted gene content in high-quality MAGs; and minimum occurrence of gene predictions in the sampled lichens. We also concluded that it would not permit inferences on absence and by extension any statistical inference on microbial occurrence or distribution of function across lichen symbioses. Even taking into consideration these limitations, our exploration of the bacterial data yielded a set of patterns that to us were unexpectedly structured and are to our knowledge novel:

1. Four family-level lineages from 2 phyla accounted for as many bacterial occurrences in lichens as all other 71 families from 16 phyla combined.

2. Among those 4 lineages, 2 network sharing motifs are visible: one in which extensive species-level sharing is evident (Acetobacteraceae and Beijerinckiaceae) and 2 in which few species are shared by more than 2 or 3 lichen metagenomes (Acidobacteriaceae and Sphingomonadaceae).

3. The 4 lineages, when screened with 16S rRNA, are found in even more lichens, with the 2 families that exhibit extensive species-level sharing occurring in 99% (Acetobacteraceae) and 97% (Beijerinckiaceae) of metagenomes, respectively.

4. The 4 lineages are recovered (as mappable MAGs; S8 Table) in the 4 data sets with more than 10 metagenomic libraries each produced for different studies in 3 different labs, as well as in more recently released data sets which we screened using 16S rRNA.

5. The high frequency of these 4 families is consistent with previous results of amplicon sequencing of lichens involving, e.g., 9 LFS genera in Alaska [71], 4 species from Colorado [72], 7 species from Antarctica [73], and 7 genera sampled in the Colombian páramo [74].

6. Some of the most frequently detected MAGs belong to bacterial genera that have either already been the focus of lichen microbiological studies or are close phylogenetic relatives of them. In our study, MAGs that cluster with LAR-1 [46,71], recently formally described as the genus *Lichenihabitans* [75,76] (Beijerinckiaceae, syn. *Lichenibacterium* [52], first validly published 8 months later [77]), were recovered in 99 metagenomes, with 1 individual MAG recovered in 52 metagenomes (Fig 2C) and accordingly shared among many lichen symbioses (S4 Fig). The second-most commonly detected bacterial "genus" that we recovered in the MAG set, tracked in GTDB as CAHJXG01, was previously known from a single MAG from a *Peltigera* lichen [30], but assembled in our study as multiple MAGs and recovered in 58 metagenomes (it could not be included in SSU rRNA screening as its SSU rRNA is absent from reference databases and not part of the MAG). Among other most frequently detected genera are *Lichenicoccus* and *Lichenicola*, both recently described genera cultured from lichens [54,78]. Another frequently occurring "genus," LMUY01, is described from a soil sample in the Czech Republic, but it was previously classified as *Acidisphaera* and subsequently segregated as a distinct clade. In a study of *Peltigera* lichens, bacteria from this genus were shown to be enriched, while being present in lichen substrata only in small amounts [79]. Other bacterial lineages previously postulated to play a metabolic role in lichens, including Acidobacteriaceae and Sphingomonadaceae, were also nearly ubiquitous in rRNA screening. Sphingomonadaceae were also recovered as MAGs in nearly half of metagenomes from lichens with cyanobacterial photobionts, echoing previous results that showed a notably greater abundance of sphingomonads in cyanolichens than in those with algal photobionts [71,74].

The annotations of MAGs from the highest frequency bacterial genera largely corroborate predictions from the last 20 years based on metaproteomics from whole lichens and characterizations of isolated bacteria. An unexpectedly common feature, only recently characterized in a lichen bacterium by Pankratov and colleagues [54,80], is the capacity for anoxygenic photosynthesis. This is consistent with the profile of AAPs, a group of bacteria that has only recently begun to gain attention for its occurrence in diverse extreme environments [55]. The capacity for methylotrophy, previously identified in bacterial annotations from a *Lobaria* lichen [53], proved to be widespread. A notable exception to the corroboration of past claims is nitrogen fixation. Despite the history of claimed nitrogen fixation by lichen bacteria, it received no support as a general phenomenon from either annotations or a targeted pan-assembly search for the *nifH* gene. The cellular sources of *nifH* sequences obtained by other researchers [11,45,46,47] are impossible to reconstruct from our data, but we cannot rule out that *nifH*-containing bacteria might occur in low quantity fractions that would be detectable by PCR, or occur in lichens not included in our data (e.g., certain tropical lichens [46]).

Cofactor metabolism stands out in the lichen microbe data as one of the areas with obvious potential for bacterial-eukaryotic cross-feeding. Most lecanoromycete fungi, and by extension most LFSs, appear to be biotin and/or thiamine auxotrophs. Some of the most frequent bacteria, for their part, appear to be biotin prototrophs, and algae have partially complete predicted pathways for biotin and thiamine; in the few metagenomes where we have at least one complete MAG of each, both algae and bacteria appear to complement the fungus. Fungal vitamin B auxotrophy is not common knowledge in lichen biology, but it is not a new discovery. Several lichen researchers documented vitamin B requirements of about a dozen LFS species in the 1950s [81,82,83,84]. Bednar [83,84] documented biotin auxotrophy in detail in the LFS of the lichen *Peltigera aphthosa*. He found that its alga, *Coccomyxa* sp., liberated large amounts of biotin *in statu symbiotico*, and proposed that this was one of the underlying syntrophies of lichen symbiosis. It is possible that the only partial completeness of biotin synthesis we predicted for algae is an annotation artifact, or that more work needs to be done to understand biotin synthesis in algae, as well as in lichen-associated bacteria. In particular, we were not able to identify in our genomes the pathway for the synthesis of pimelate moiety, one of the biotin precursors. Enzymes responsible for this step are highly diverse [85], with alpha-proteobacteria in particular having a non-standard pimelate synthesis pathway [86]. Since this made large-scale screening challenging, in our analysis we focused instead on the modules M00123 and M00950, which contain enzymes that catalyze biotin ring assembly and are generally conserved [86].

Cobalamin synthesis is predicted for many of our high frequency bacteria and has been hypothesized to sustain cobalamin-auxotrophic algae in vivo [9]. However, none of the algal MAGs we queried unambiguously require cobalamin. Instead, they possess both *metE* and *metH*, suggesting that, if they function similarly to *Chlamydomonas reinhardtii* [61], lichen algae might be facultatively dependent on cobalamin for methionine synthesis.

## What do our results mean for research on lichen-associated bacteria?

There is a long history of christening lichen bacteria as "symbionts," from the 1920s and 1930s [23,44,87] up to the present [18,53,58,71]. The term "symbiosis" originally denoted the stable co-occurrence or "living together" of different species, regardless of whether the relationship is mutualistic, commensalistic or even parasitic [2], and one can surmise that this has been the sense in which the term has been applied in the lichen context. In practice, however, "symbiosis" has increasingly come to be applied to one segment of a wider spectrum of organismal co-occurrence phenomena in which the players perform some kind of required function. The

remaining part of this spectrum is usually referred to as the microbiome, though it should be noted that many cross-feeding relationships (syntrophies) have been described from microbiome data [88] without necessarily referring to them as symbioses. Against this background, although bacteria are physiologically active and have been shown to produce multipurpose metabolites in the lichen milieu (reviewed by [13]), to our knowledge no studies have shown any metabolic function that locks the now-canonical fungal and photobiont symbionts into a dependency on bacteria, or vice versa.

Our analyses of the high-frequency bacterial fraction inject new data into this discussion. The high frequency occurrence at the family level of Acetobacteraceae and Beijerinckiaceae in lichen symbioses is arguably consistent with the broadest, classical definition of symbionts. Beyond this, however, our results are insufficient to advance a narrower, function-based case for bacterial symbiosis in lichens. Our predictions of biotin prototrophy in some high-frequency lichen bacteria and corresponding auxotrophy in lichen fungi suggests the potential of fungal dependency on these bacteria, but much will depend on mapping the constellations in which prototrophs and auxotrophs co-occur (or do not co-occur). Certainty regarding the fungal sourcing of biotin will require targeted studies that explore a range of possibilities. An alternative scenario that has already been proposed, albeit without considering a possible role for bacteria, is that algae cover the biotin needs of their fungal partners [83]. It is also possible that biotin-prototrophic bacteria occur in such small amounts that their contribution to eukaryotic metabolism is physiologically negligible. An additional and not mutually exclusive possibility is that biotin-prototrophic bacteria play a role in the poorly studied period of the fungal life cycle before spores and young mycelia pair with a photobiont. We note however that none of the metagenomes included in this survey were sampled from this life stage and we have no evidence that biotin-producing bacteria are associated with the LFS outside of lichens. In any case, if experiments confirm widespread LFS biotin auxotrophy, provisioning from any exogenous source would constitute an important and hitherto largely overlooked metabolic interaction in lichen symbioses that requires further study.

Several lines of evidence suggest that some of the high frequency bacterial lineages, for their part, also depend on metabolic subsidies or feed off the lichen. Most of the annotated bacteria are predicted thiamine auxotrophs or possess only a thiamine scavenging pathway, and could conceivably use algal or fungal thiamine or precursors; yet others are predicted biotin auxotrophs. Carbohydrate transporter and CAZyme annotations suggest that some bacteria tap into the abundant polyols of lichen symbioses and, especially for Acidobacteriaceae, may be involved in degradation of eukaryotic cell walls, as has been suggested before [89,90]. Taken together, these predictions could support the interpretation that some bacteria are commensals that make a living off the lichen environment.

Our approach to analyzing only the most frequently occurring bacteria with the highest quality MAGs yielded a narrow set of taxa. We consider the probability that these are contaminations to be minimal, as they closely mirror those found in previous studies, e.g., using metabarcoding, were recovered across multiple metagenome sequencing projects, and exhibit little to no overlap with the documented "kitome" taken up from laboratory sample handling [91]. Beyond the high frequency bacterial taxa we analyze here, we cannot rule out that lineages that occur at lower frequencies in our data sets, or were not fully recovered as high-quality MAGs, may ultimately deserve further study in the context of individual lichen "species." An example is Actinobacteriota, which we recovered as MAGs at low frequency, but which have been highlighted, e.g., by Gonzales and colleagues [92], cultured from some lichens ([93], also see Fig 2C), and postulated to be rich in bioactive compounds [13]. Distinguishing a "lichen-relevant" bacterial fraction from contaminations becomes an increasingly non-trivial problem at lower occurrence frequencies, and at least 3 different solutions have been proposed to address

this. One has been to wash lichen thalli with water, ethanol, and bleach prior to downstream use, a treatment borrowed from the study of endophytes [94,95]. A downside of this treatment is that the lichen exopolysaccharide layer is of variable composition [18] and thallus washes may dislodge more than just surficial microbes, based on our experience with warm water [31]. Another downside, in our view, is that it assumes, to our knowledge without testing, that whatever microbial fraction is lost is not metabolically relevant. A second approach is to compare microbial composition on lichens with those of surrounding environments. We think that much stands to be learned from environmental comparisons about microbial ecology, but here, too, we are not convinced that we know enough about lichen microbiology to know a priori that occurrence in a non-lichen environment means that a microbe can be ruled out as participating in the lichen system. In fact, it is well known that canonical eukaryotic lichen photobionts are commonly sampled in the environment (reviewed by [96]). This need not be disqualifying for lichen symbiosis if specific roles such as carbon fixation or vitamin provisioning are filled under an "It's the Song, Not The Singer" model [97] in which the overall lifestyle of the microbe is thought to be less important than that it is compatible with—and performs a function in—the system, when present. A third approach, similar to our approach here of identifying high frequency bacterial species, has been to resample individual lichen symbioses from spatially or temporally independent sites by metagenomics or metagenomics coupled with custom PCR screening, as has been done for lichens involving the LFSs *Alectoria sarmentosa*, *Bryoria fremontii*, and *Letharia vulpina* ("secondary screening" sensu [9]). This approach however also has the downside that it emphasizes taxonomic identity over function and compatibility.

## Conclusions

Lichens are notoriously recalcitrant to experimentation in the lab and their component symbionts are extraordinarily slow-growing, when they can be cultured at all. The ability to extract MAGs from metagenomes has been a game-changer for lichen research, but many questions remain. The survey of metagenomic data is an opportunity to take stock of their organismal content, evaluate their utility for microbial and symbiont detection, and extract a substantial body of new data. Novelties in microbial occurrence patterns and gene annotations aside, one of the more striking results from our survey is just how few metagenomes (16 of 437, after deduplication) yielded anything close to a set of MAGs that would allow even a cursory assessment of microbial interactions between at least one fungus, one photobiont, and one bacterium. Even in these metagenomes, the majority of MAGs were not of sufficient quality to use for comparison, and none included MAGs of basidiomycete fungi that widely occur as yeasts, meaning that even in the most deeply sequenced metagenomes, the metabolic predictions are potentially only a teaser of a larger story.

A leading factor limiting MAG recovery in metagenomes is sequencing depth. Published lichen metagenomes span 6 orders of magnitude of sequencing coverage, and skew low. Consequently, the data cannot be used at all for estimates of absence, and organismal occurrence frequencies are likely underestimates across the board. This is clearly visible for known photobionts and can be expected to disproportionately affect eukaryotes, with their larger and more complex genomes. Deeper sequencing will likely alleviate, though not fully resolve, this problem. Some of the problems of, e.g., MAG recovery and rRNA detection arise from the application of short read-based DNA sequencing on samples with specific biological attributes, such as the co-occurrence in a single sample of highly similar DNA sequences by virtue of multiple chromosome sets or multiple genotypes. As long read sequencing becomes more tractable, some of these problems may be resolved, such as removing issues associated with assembly

and ambiguities introduced from binning approaches. Techniques such as dynamic adaptive sequencing [98] may further facilitate the sequencing of relatively simple communities, to obtain genomic sequences of even low abundance members of the community.

## Methods

### Ethics statement

Newly collected samples acquired at Cardinal Divide, Mountain Park and Kootenay Plains (Alberta, Canada) were collected under a collection permit issued on 09.07.2018 by Alberta Environment and Parks (permit number: N/A_2). No other areas required special permitting. The existing herbarium specimens from which all other newly sequenced metagenomes were derived (S2 Table) are deposited in the collections of Uppsala University, Sweden (UPS), the University of Bergen, Norway (BG), the University of Graz, Austria (GZU) and the Universidad Nacional de Comahue, Argentina (BCRU).

### Data set construction

We analyzed a total of 480 lichen metagenomes. Data were obtained from ENA (S1 Table) and complemented with 24 metagenomes that were newly generated for this study. In the early stages of our analysis, we removed 43 metagenomes from the data set as they were identified as duplicates (S3 Table). To identify such metagenomes, we used sourmash v4.2.2 [99]; we used the "sourmash sketch" module to compute signatures and the "sourmash module" to compare individual metagenomes. This way, we identified 43 pairs of identical metagenomes; one metagenome from each pair was subsequently removed from the analysis. To generate new metagenomes, we collected lichen samples, froze them at –80˚C, and pulverized them using a TissueLyser II (Qiagen). We extracted DNA from the samples with DNAEasy Plant Mini Kit (Qiagen) and prepared metagenomic libraries. The libraries were sequenced on different Illumina HiSeq platforms to paired-end reads. The details on the procedure, including voucher information, library prep, and sequencing are given in S2 Table.

### Initial steps of metagenomic analysis

We started by assembling each metagenome individually and extracting MAGs from them. The metagenomic libraries were filtered using fastp [100] to remove adapters and low-quality bases and the READ_QC module of the metaWRAP pipeline v.1.2 [101] to remove human contamination. The filtered data were assembled with metaSPAdes [102]. Individual assemblies were binned using CONCOCT [103] and metaBAT2 [104]. To refine prokaryotic MAGs, we used the *binrefine* module of the metaWRAP pipeline and then evaluated all bins with CheckM v1.1.3 [105]. Next, we selected all bins that passed the QS50 threshold [106] and dereplicated them using dRep v3 [107] at 95% ANI and 30% AF (alignment fraction) thresholds in order to obtain species-level representatives. We did not undertake co-assembly as this can result in the merging of closely related species and we would have lost the relationship between sample and genome. Furthermore, given the number of metagenomes involved it would have been prohibitively expensive computationally to assemble all of the data and the diversity of the lichen genomic composition was not understood to facilitate selective sample co-assembly.

We obtained taxonomic assignments for prokaryotic MAGs using GTDB-Tk v1.5.0 [108], a tool based on the Genome Taxonomy DataBase (GTDB) [41]. In the text, we refer to the GTDB-defined lineages CAIMSN01 and VCDI0 as *Lichenicola* and *Lichenicoccus*, respectively, as they are referenced in the later GTDB release (R214). We generated a phylogenomic tree for all prokaryotic MAGs that passed the QS50 threshold. In the tree, we also included reference

genomes from bacteria previously isolated from lichens (S5 Table). We used the marker gene alignment produced by GTDB-Tk (concatenated alignment of 120 loci). We generated the tree with IQ-TREE [109] using the model finder (selected model: LG+F+R10) and 1,000 bootstraps.

Eukaryotic MAGs were identified and refined with EukCC v2 [110]. Bins with a quality score of at least 50 were dereplicated with dRep at 2 levels: first, on the level of the individual binned metagenome (with the 99% ANI threshold), and second, on the level of the whole data set, where bins from all metagenomes were dereplicated at 95% ANI and 40% AF to create species-representative MAGs. For each MAG, we calculated the EukCC v2 and BUSCO v5 [111] quality scores. To analyze the relationship between sequencing depth and recovery of MAGs of the LFS and photobionts, we used the pre-dereplication MAG set.

To obtain preliminary taxonomy annotations for eukaryotic MAGs, we used BAT (CAT v5.2.3, database version: 20210107 [112]), which predicts taxonomy based on searching predicted genes against the NCBI database. These taxonomic assignments were refined using phylogenomics. We separated all eukaryotic MAGs into 2 groups: fungal and algal MAGs. To both groups we added reference genomes (S5 Table). To compute the fungal phylogenomic tree, we used the Phylociraptor pipeline v0.9.6 (https://github.com/reslp/phylociraptor), executing the following steps. In each genome, we identified BUSCO universal single copy orthologs shared by at least 10% genomes in the set. We aligned the sequences and trimmed the alignments. From these data, we produced 2 trees: a coalescence tree that was reconstructed by ASTRAL v5.7.1 [113] from the individual gene trees (produced by IQ-TREE v2.0.7 [109]), and a tree calculated from a concatenated alignment using IQ-TREE. We compared the coalescence and the concatenated phylogenies and reconciled detected discrepancies using signal base approximation in favor of the concatenated phylogeny. The algal phylogenomic tree was produced in the same way. The final trees were based on 1,296 genes in algae and 709 genes in fungi.

## Occurrence analysis

To map MAG occurrence across the metagenomes, we aligned reads from all metagenomes against all MAGs using BWA-mem [114]. Next, we filtered the alignments using SAMtools [115] to remove secondary alignments. All MAGs that were at least 50% covered in a given metagenome were counted as present. Using these data, we constructed an occurrence matrix of MAGs in metagenomes. To estimate the depth of coverage of MAGs, we used the number of reads aligned to the MAG, multiplied by the read length and divided by the total length of the contigs assigned to the MAG.

In each metagenome, we identified the MAG of the LFS. To do that, we manually inspected all fungal MAGs present in a metagenome. If only one fungal MAG was present, it was labeled as putative LFS. If multiple fungal MAGs were present, we selected one as the LFS MAG based on its position on the tree and the one with the highest depth of coverage, since the MAG of the main, most abundant LFS is expected to have greater depth of coverage than a MAG from a different fungus. To confirm the LFS assignments, we checked that the MAG placement on the phylogenomic tree is consistent with the taxonomic assignment provided by the original data submitters in the NCBI metadata. If these did not match, we excluded these metagenomes from Dataset 2 as potentially derived from misidentified samples. In total, we identified 23 inconsistencies. In 3 metagenomes (SRR14722059, SRR14722135, and SRR14722098), the inconsistency was easily resolved by correcting the putative LFS assignment and giving it to a different fungal MAG present in the metagenome. In the additional 2 metagenomes (SRR14722289 and SRR14721950), NCBI metadata had inconsistencies within itself: the lichen name in the "organism" field did not match the name in the "library name." Our taxonomic placement agreed with the latter, and therefore we suspected that the "organism" field was

filled in incorrectly during data uploading. We retained these 2 samples, correcting their names to their "library names." A total of 18 metagenomes presented inconsistencies that could not be resolved and were thus excluded (S6 Table).

All retained metagenomes were used with the name of the LFS as assigned in the original publication, with one exception, *Lecidea scabridula* Hedl. was described from Sweden by Hedlund [116] and is a distinct fungal taxon engaged in a biofilm-like lichen symbiosis that has appeared in taxonomic works for that region and was reported as new to North America by Spribille and Björk [117]. As the latter authors acknowledged, *Lecidea scabridula* Hedl. is an illegitimate name as it is a later homonym of *Lecidea scabridula* Müll. Arg; accordingly, the species represented by *Lecidea scabridula* Hedl. currently has no legitimate name. A nomenclatural and taxonomic revision has been in preparation for some time (T. Spribille and M. Svensson, in prep.) but remains uncompleted as of this writing. Based on information available at the time to TS, who provided the sample, Resl and colleagues [36] applied the name *Bachmanniomyces* sp. S44760 to a sample corresponding to *L. scabridula* from Alberta, Canada, as *Bachmanniomyces* was thought to represent the correct generic placement of the species. We however no longer consider this to be the case. To avoid any ambiguity, or the perception that the species is a lichenicolous fungus as other members of the genus *Bachmanniomyces*, we have reverted in the present paper to applying its de facto name *Lecidea scabridula* Hedl. to the metagenomic library sequenced from the sample T1894.

### SSU rRNA gene-based screening

We searched metagenomic assemblies and raw, unassembled metagenomic data for the presence of the SSU rRNA gene of several lineages. This process consisted of 2 steps: the detection of bacterial SSU rRNA and eukaryotic SSU rRNA sequences, and their taxonomic assignment. These sequences were used for 2 reasons: first, they are the marker loci most frequently used for taxonomic profiling, and second, they tend to be present in multiple copies in a genome [118] and therefore have better chances of being recovered in a shallow metagenome. For the first step, we used Metaxa2 [119], a tool that uses an HMM-based searching algorithm followed by a taxonomic assignment via BLAST search against a SILVA database. For eukaryotic lineages, taxonomic placement was done through Metaxa2 as well. For bacteria, we used SSU rRNA sequences extracted by Metaxa2, to which we assigned taxonomic positions with IDTAXA [120], which allowed us to use taxonomy consistent with GTDB. Only metagenomes from the Dataset 1 were included in this analysis ($n$ = 375; Fig 1B).

### Identifying most frequent bacterial groups

We ranked bacterial groups based on their frequency, defined as the total number of occurrences across the data set. We summarized frequency on 4 taxonomic levels: species-level lineage, genus, family, and order. For the species-level lineages, we simply counted how many metagenomes they were detected in. For the higher taxonomic levels, we summed all occurrences of all lineages assigned to that group. If a MAG did not have a genus level assignment, we used its family-level annotations (e.g., Acetobacteraceae gen. sp.). In addition, we ranked higher-level taxonomic groups based on how many species-level lineages from this group were detected. When calculating the percentage of metagenomes, a given lineage was detected in, we only included metagenomes from the Dataset 1 (Fig 1B).

### Co-occurrence analysis

To explore how lineages co-occur within lichen samples, we built co-occurrence network graphs (S4 Fig), using the occurrence matrix. We defined co-occurrence as an instance of 2

lineages occurring together in one metagenome. For this analysis, we focused on the groups that are known to stably occur in lichens (algae, Cyanobacteria, and Cystobasidiomycetes and Tremellomycetes fungi) and on the most frequent bacterial groups (most frequent genera of Acetobacteraceae, Beijerinckiaceae, and Acidobacteriaceae). Only metagenomes that yielded an LFS MAG (*n* = 330; Dataset 2) were included in this analysis.

## Functional analysis: Bacteria

We annotated all bacterial MAGs using PROKKA v1.13 [121]. Predicted proteins in lichen metagenomes were clustered to the MGnify protein database [122] using the Linclust algorithm in mmseqs2 v13.45111 [123] at 90% coverage and 90% sequence identity. Next, we selected the MAGs of the most frequent lineages and annotated them in depth. To select the MAGs, we first ranked all bacterial genera based on the number of occurrences. For the MAGs that did not have a genus level assignment, we used family-level annotations. Next, we selected the MAGs assigned to the top 13 genera, and among them retained only MAGs with a completeness score above 95% and contamination score below 10%, as estimated by CheckM.

For the selected MAGs, we obtained functional annotations. We annotated predicted proteins against KEGG Orthology Database [124] using KofamScan [125]. The resulting KEGG ortholog (KO) assignments were used to estimate KEGG module completeness using ggkegg [126] with the KEGG module definitions outlined in the KEGG MODULE Database (www.genome.jp/kegg/module.html). We focused on several metabolic traits, which we expected to be relevant to the symbiosis:

1. Carbon metabolism. We screened the MAGs for known carbon fixation pathways, including Calvin–Bensen path (KEGG module M00165), and 6 alternative pathways (see [55]): reductive citrate cycle (M00173), 3-hydroxypropionate bi-cycle (M00376), hydroxypropionate-hydroxybutylate cycle (M00375), dicarboxylate-hydroxybutyrate cycle (M00374), Wood–Ljungdahl pathway (M00377), or the phosphate acetyltransferase-acetate kinase pathway (M00579). We also searched for the genes related to C1 metabolism: methanol dehydrogenase (KEGG family K23995) and methane monooxygenase (K10946 and K16157). In Cyanobacteria, we were not able to find one of the enzymes of the Calvin–Bensen path, sedoheptulose 1,7-bisphosphatase. However, since its function can be performed in Cyanobacteria by fructose-1,6-bisphosphatase II [127], we still show the Calvin–Bensen path as complete.

2. Nitrogen metabolism. We screened the MAGs for the presence of nitrogenase *NifH* (K02588; involved in nitrogen fixation) and urease (EC 3.5.1.5).

3. Photosynthesis. We screened the MAGs for the proteins of Anoxygenic Photosystem II (M00597 and M00165; *pufABCML-puhA*), and for biosynthetic pathways for the photosynthetic pigments: bacteriochlorophyll (K04035, K04037, K04038, K04039, K11333, K11334, K11335, K11336, K11337, K04040, K10960; *AcsF, ChlBNL, BchCFGPXYZ*), and carotenoids (K02291, K10027, K09844, K09844, K09845, K09846).

4. Transport systems. We annotated the following transport systems and transporters: sorbitol/mannitol transporter (K10227, K10228, K10229, K10111; *SmoEFGK*), urea transporter (K11959, K11960, K11961, K11962, K11963; *urtABCDE*), erythritol transporter (K17202, K17203, K17204; *EryEFG*), xylitol transporter (K17205, K17206, K17207; *XltABC*), inositol transporter (K17208, K17209, K17210; *IatAP-IbpA*), glycerol transporter (K17321 K17322, K17323, K17324, 17325; *GlpPQSTV*), fucose transporter (K02429; *FucP*), glycerol aquaporin transporter (K02440; *GLPF*), glucitol/sorbitol transporter (K02781, K02782, K02783;

*SrlABE*), ammonium transporter (K03320; *Amt*), ribose transporter (K10439, K10440, K10441; *RbcABC*), xylose transporter (K10543, K10544, K10545; *XylFGH*), multiple sugar transporter (K10546, K10547, K10548; *ChvE-GguAB*), fructose transporter (K10552, K10553, K10554; *FrcABC*), arabinose transporter (K10537, K10538, K10539; *AraFGH*), branched-chain amino acid transporter (K01999, K01997, K01998, K01995, K01996; *LivGFHKM*), L-amino acid transporter (K09969, K09970, K09971, K09972; *AapJMPQ*), glutamate transporter (K10001, K10002, K10003, K10004; *GltIJKL*), capsular transporter (K10107, K09688, K09689; *KpsMTE*).

5. Cofactors. We searched for the biosynthetic pathways of the following cofactors: biotin (M00123, M00577, and M00950), thiamine (M00899; thiamine salvage pathway), cobalamin (M00122), and riboflavin (M00125).

We used several tools to annotate groups of genes that are potentially informative to the symbiotic lifestyle. We used following tools: run_dbcan (standalone tool of dbcan2, v3.0.2, https://github.com/linnabrown/run_dbcan) for annotations of Carbohydrate-Active EnZymes (CAZymes), FeGenie [51] for the genes related to iron metabolism. We used Sanntis [128] and antiSMASH v6.1.0 [129] for biosynthetic gene clusters. Both tools predict BGCs from the genome sequences and annotate predicted BGCs by comparing them to the MiBIG database. We screened these annotations for 2 groups of interest: BGCs potentially producing carotenoids and BGCs potentially producing extracellular polysaccharides. While analyzing the antiSMASH results, we used only BGCs that had significant hits to BGCs from the MiBIG database according to the outputs of the KnownClusterBlast module. While analyzing the SanntiS results, we retained only annotations that had Jaccard distance score below 0.7. Our SanntiS results should be considered as preliminary, since they are based entirely on sequence similarity, and we did not further validate the annotations by taking into account the position of the domains or their copy number.

To account for the possibility that *NifH* is present on a plasmid and therefore failed to be included in the MAG during binning, we further searched *NifH* across all contigs of all metagenomic assemblies, using tBLASTn [130] and a *NifH* sequence from NCBI as a query (S14 Table). We extracted all hits with e-value below 1e-50 and checked their taxonomy using reciprocal blast search against the NCBI database using getLCA (https://github.com/frederikseersholm/getLCA). All hits were assigned to Cyanobacteria, with 1 exception: 1 metagenome (SRR14722280) contained a Hyphomicrobiales (Rhizobiales) hit on a low-coverage contig not assigned to any MAG isolated from this metagenome.

## Functional analysis: Eukaryotes

We annotated high-quality eukaryotic MAGs (≥95% complete and ≤10% contaminated), using Funannotate v1.8.15 [131]. Briefly, each MAG was cleaned and sorted and used for gene prediction with Genemark-ES v4.62 [132], Augustus v3.3.2 [133], CodingQuarry v2.0 [134], GlimmerHMM v3.0.4 [135], and SNAP 2006-07-28 [136]. We annotated predicted proteins using InterProScan v5.42–78.0 [137] and KofamScan [125]. Resulting KEGG annotations were processed as described above.

To establish whether the algae in our data set are cobalamin auxotrophs, we followed Croft and colleagues [61] and screened the algal MAGs for a cobalamin-dependent methionine synthase *MetH* and another gene performing the same function—cobalamin-independent *MetE*. First, we selected algal MAGs with completeness >90% according to EukCC (*n* = 19). Next, we screened them using tBLASTn [130] and 2 protein sequences as a query (BAU71143.1 for *MetH* and BAU71146.1 for *MetE*). We confirmed the identity of the resulting hits by a reciprocal search against the NCBI database.

## Loss of function in Hyphomicrobiales MAGs

Hyphomicrobiales (Rhizobiales) MAGs from our data set lacked several functions typical for bacteria from this order. To put these MAGs into the evolutionary context, we assembled a data set that included 518 previously published genomes across the whole order [138] (S20 Table) and a genome of *Rhodobacter* (GCF_009908265.2), which served as an outgroup. Using GTDB-Tk, we identified and aligned 120 marker genes. From this alignment, we generated a phylogenomic tree using IQ-TREE v2.1.2 [109]. We used tblastn to screen all Hyphomicrobiales MAGs for the same genes related to nitrogen fixation, methanotrophy, and methylotrophy (S14 Table). For the genomes from GenBank, we confirmed that the tblastn results were consistent with the protein annotations available at NCBI.

## Screening of newly published metagenomic data

Our initial data set was built before October, 2021. To confirm that our results were consistent with more recently published metagenomes, we additionally reran our occurrence analysis on these new raw sequence reads. We queried the SRA using the rentrez package in R [139] with search terms chosen to capture all reads associated with lichens. Our terms included "Pezizomycotina" and "lichen metagenome" in the lichen taxonomy, or the terms "lichen" or "lichens" in any of the searchable fields. To reduce duplicating records previously found, we only searched for records submitted after October 1, 2021. Finally, to make these results comparable to our earlier assembled data set, we included only records that used a whole genome sequencing strategy and were sequenced on Illumina machines. This produced a list of 7,324 metagenomes potentially relevant for our analysis. We manually checked SRA metadata for each entrance to confirm that the metagenomes originated from lichen samples. Since cultured or excessively cleaned lichens may have lost their full diversity, we also confirmed via literature associated with the records that the lichen thallus was collected in the field and was not cleaned beyond removing debris. This reduced the candidate data set to 243 metagenomes generated since October 1, 2021. We confirmed no duplicate metagenomes were present using sourmash [99], comparing both the new and previously used metagenomes. No duplicate metagenomes were found in the final set of raw reads. To detect the presence of bacteria in the 243 new metagenomes, taxonomy was assigned to raw reads using the Metaxa2 [119] and IDTAXA [120] approach described in our SSU rRNA gene-based screening methods.

## Data handling and visualization

Custom scripts used for data analysis and visualization were written in R v4.1.0 [140], using the following libraries: dplyr v1.0.8 [141], tidyr v1.2.0 [142], scales v1.1.1 [143], for data handling; ggplot2 v3.3.5 [144], ape v5.0 [145], phangorn v2.8.1 [146], phytools v1.0–3 [147], circlize v0.4.14 [148], qgraph v1.9.2 [149], treeio v1.16.2 [150], DECIPHER v2.14.0 [151], for data visualization. For visualizing phylogenetic trees, we also used iTOL [152].

## Data and code availability

The sequencing data in this project are submitted to ENA: de novo generated raw data (study accession PRJEB59037), metagenomic assemblies (PRJEB72384, PRJEB72386-PRJEB72404, PRJEB72498- PRJEB72501), and MAGs (PRJEB77567). Phylogenomic trees in Newick format are available at FigShare (10.6084/m9.figshare.27054937). Custom scripts used for data analysis

and visualization are available on GitHub (https://github.com/Spribille-lab/2024-Microbial-occurrence-in-lichen-metagenomes) and FigShare (10.6084/m9.figshare.27054937).

## Supporting information

**S1 Fig. Dot histogram of lichen metagenomes arranged by sequencing depth.** Each metagenome is shown with a dot; the dots are colored according to the taxonomy of the lichen fungal symbiont (LFS). The dots are arranged on the x-axis based on the sequencing depth (bp) and are "stacked" on top of each other. The data underlying this figure can be found in S1 Table.
(TIF)

**S2 Fig. Sampling locations for the metagenomic data used in the analysis and the geographic distributions of most frequent bacterial genera.** Each dot represents a sample used for metagenomic sequencing. The basemap shapefile was taken from the rnaturalearthdata library (v0.1.0) and belongs to public domain (https://www.naturalearthdata.com/).
(TIF)

**S3 Fig. Geographic distribution of selected groups of symbionts, based on 3 methods of screening: presence of MAGs, presence of SSU rRNA gene in the metagenomic assemblies, and presence of the SSU rRNA in the raw, unassembled reads.** Each dot represents a sample used for metagenomic sequencing. Here are shown data on the 4 most frequent bacterial families, on the genus *Lichenihabitans*, and on the 3 eukaryotic lineages known to be stably associated with lichens. The basemap shapefile was taken from the rnaturalearthdata library (v0.1.0) and belongs to public domain (https://www.naturalearthdata.com/).
(TIF)

**S4 Fig. Co-occurrence networks of lichen symbionts based on presence-absence of MAGs in each metagenome.** Each node is a MAG, and edges represent the co-occurrence of MAGs within 1 metagenome; the thicker the edge, the more often 2 MAGs co-occur. Nodes are colored based on the taxonomy and function of the symbiont; in each network, yellow nodes represent MAGs of the LFS (lichen fungal symbiont). Only data on metagenomes that yielded an LFS MAG are shown. (A) Co-occurrence of LFSs, other known eukaryotic symbionts, and Cyanobacteria. (B) Co-occurrence of LFSs and *Lichenihabitans*. (C) Co-occurrence of LFSs and most frequent genera of Acetobacteraceae. (D) Co-occurrence of LFSs and most frequent genera of Sphingomonadaceae. (E) Co-occurrence of LFSs and most frequent genera of Acidobacteriaceae. The data underlying this figure can be found in S1 Data.
(TIFF)

**S5 Fig. Maximum likelihood phylogenetic tree of Hyphomicrobiales (Rhizobiales).** The tree includes published genomes of Hyphomicrobiales and the Hyphomicrobiales MAGs derived from the lichen metagenomes (indicated in red). We generated the alignment of 120 marker genes using GTDB-Tk and calculated the tree using IQ-TREE. The color represents family-level taxonomic assignment. We used tblastn to search the genomes for key genes involved in nitrogen fixation and C1 metabolism. The presence of these genes is indicated with symbols. The full-size version of the tree in the Newick format is available at FigShare (10.6084/m9.figshare.27054937).
(TIF)

**S6 Fig. Recovery of MAGs of the main 2 symbionts as a function of sequencing depth.** These graphs are based on the pre-dereplication set of MAGs, each dot represents a metagenome and is positioned based on its sequencing depth and on whether it contained MAGs

assigned to one or both of the 2 main partners: (A) the LFS (lichen fungal symbiont); (B) the photobiont partner; (C) both the LFS and the photobiont partner. The data underlying this figure can be found in S1 Data.
(TIFF)

**S7 Fig. Relative abundances of symbionts in lichen metagenomes.** The relative abundances were calculated by dividing the coverage depth of the symbiont MAG by the coverage of the LFS MAG. Here are shown data on the 13 most frequent bacterial genera and the eukaryotes known to be stably associated with lichens. The red line shows 1:1 ratio, where the symbiont is estimated to have the same cellular abundance as the main fungal symbiont. The boxplot elements are defined as: center line, median; box limits, 25th and 75th percentiles; whiskers, 1.5× interquartile range. The data underlying this figure can be found in S1 Data.
(TIFF)

**S1 Data. Excel spreadsheet containing, in separate sheets for each figure, the underlying data used for Figs 3, 4B, 5A, 5B, 5C, 6, 7, S4, S6, and S7.**
(XLSX)

**S1 File. KEGG module completeness for prokaryotic MAGs selected for analysis.**
(PDF)

**S2 File. KEGG module completeness for eukaryotic MAGs selected for analysis.**
(PDF)

**S1 Table. Metagenomic data used in the analysis.**
(XLSX)

**S2 Table. Details on the metagenomes generated de novo for this study.**
(XLSX)

**S3 Table. Pairs of identical metagenomes (sourmash similarity = 1) identified during analysis.** Using sourmash, we identified the pairs of identical metagenomes, potentially deriving from one data set having been submitted to NCBI twice under different accession numbers. This table contains all such pairs and shows which of the pair was retained for the further analysis.
(XLSX)

**S4 Table. Metagenome-assembled genomes generated from the lichen metagenomic data.** For prokaryotic MAGs, contamination and completeness scores are based on the CheckM results, and taxonomic assignments are based on GTDB. For eukaryotes, contamination and completeness scores are based on the EukCC results, and taxonomic assignments are based on the BAT assignments corrected via phylogenomic analysis. Number of occurrences represents the number of metagenomes this MAG was detected in.
(XLSX)

**S5 Table. List of reference genomes used for construction of the phylogenomic trees.**
(XLSX)

**S6 Table. Metagenomes derived from potentially misidentified samples.** Metagenomes in this list had inconsistencies between their metadata and the taxonomy of the LFS as estimated from the phylogenomic tree. These metagenomes were excluded from the occurrence analysis.
(XLSX)

**S7 Table. Most frequent bacterial lineages (on the family and genus level) in various groups of lichen metagenomes.** Bacterial lineages were ranked based on the total number of

their occurrences. For each lichen group here we give top-ten bacterial taxa.
(XLSX)

**S8 Table. Occurrence of key bacterial groups in metagenomes originating from different studies.** For 13 most common bacterial genus-level lineages and 4 bacterial families, we are showing the number of metagenomes in which they were detected in as MAGs, split by the source of the metagenomic data (the study it was reported in). The total number of metagenomes that come from each study and which were included in the Dataset 1 are shown at the bottom of the table.
(XLSX)

**S9 Table. Number of MAGs from different categories in each metagenome.** The MAG categories were based on a combination of taxonomy and function. For the metagenomes that were identified as potentially derived from misidentified samples, we assigned their LFS to a separate category "LFS misassigned."
(XLSX)

**S10 Table. Bacterial families ranked on their prevalence in lichen metagenomes, as detected via screening of rRNA genes in metagenomic assemblies and unassembled reads.** Prevalence is calculated using only metagenomes from the Dataset 1 ($n = 375$).
(XLSX)

**S11 Table. Details on additional metagenomes from NCBI's SRA.**
(XLSX)

**S12 Table. Prevalence of key taxa in additionally screened metagenomes.**
(XLSX)

**S13 Table. MAGs selected for the in depth functional annotation.** To select these MAGs, we first calculated the number of occurrence per genus, and selected the 13 top genera. Next, we selected among them the MAGs with >95% completeness and <5% contamination, according to CheckM.
(XLSX)

**S14 Table. Sequences used as a blast query for the screening genomes for the signs of nitrogen fixation, methanotrophy, and methylotrophy.**
(XLSX)

**S15 Table. Number of iron-related genes annotated with FeGenie.**
(XLSX)

**S16 Table. Median number of CAZymes assigned to different classes per MAG for the most frequent bacterial genera.**
(XLSX)

**S17 Table. Number of CAZymes assigned to different CAZy families in the 63 MAGs selected for in-depth annotation.**
(XLSX)

**S18 Table. Biosynthetic Gene Clusters (BGC) predicted for the selected 63 bacterial MAGs.** We used SanntiS to predict the BGCs and to annotate them by identifying the most similar BGC included in the MiBIG database. Only the hits with <0.7 Jaccard distance are listed in this table. BGCs that are similar to known clusters producing exopolysaccharides are

indicated in bold.
(XLSX)

**S19 Table. Presence/absence of cobalamin-dependent MetH and cobalamin-independent MetE in the high-quality algal genomes (completeness >90%, contamination <10%, according to EukCC).**
(XLSX)

**S20 Table. Reference genomes from NCBI used for the Hyphomicrobiales phylogenomic tree.** Taxon sampling followed Volpiano and colleagues [138]. Taxonomic classification was done using GTDB-Tk.
(XLSX)

## Acknowledgments

Special thanks go to Piotr Łukasik and John McCutcheon for support to GT and SG during a visit to the University of Montana. Thanks also to Maria Chuvochina (University of Queensland) for clarifying the nomenclatural priority of *Lichenihabitans*.

## Author Contributions

**Conceptualization:** Gulnara Tagirdzhanova, Lisa Y. Stein, Robert D. Finn, Toby Spribille.

**Data curation:** Gulnara Tagirdzhanova, Ellen S. Cameron, Andrew T. Cook, Spencer Goyette, Veera Tuovinen Nogerius, Alfredo Passo, Helmut Mayrhofer, Håkon Holien, Tor Tønsberg, Robert D. Finn.

**Formal analysis:** Gulnara Tagirdzhanova, Paul Saary, Ellen S. Cameron, Carmen C. G. Allen, Arkadiy I. Garber, David Díaz Escandón, Andrew T. Cook, Spencer Goyette, Lisa Y. Stein, Toby Spribille.

**Funding acquisition:** Toby Spribille.

**Investigation:** Gulnara Tagirdzhanova, Paul Saary, Ellen S. Cameron, David Díaz Escandón, Veera Tuovinen Nogerius, Alfredo Passo, Helmut Mayrhofer, Håkon Holien, Tor Tønsberg, Robert D. Finn, Toby Spribille.

**Methodology:** Gulnara Tagirdzhanova, Paul Saary, Ellen S. Cameron, David Díaz Escandón, Spencer Goyette, Robert D. Finn, Toby Spribille.

**Project administration:** Toby Spribille.

**Resources:** Toby Spribille.

**Software:** Paul Saary, Ellen S. Cameron, Toby Spribille.

**Supervision:** Lisa Y. Stein, Robert D. Finn, Toby Spribille.

**Validation:** Gulnara Tagirdzhanova, Toby Spribille.

**Visualization:** Gulnara Tagirdzhanova, Paul Saary, Ellen S. Cameron.

**Writing – original draft:** Gulnara Tagirdzhanova, Toby Spribille.

**Writing – review & editing:** Gulnara Tagirdzhanova, Paul Saary, Ellen S. Cameron, Carmen C. G. Allen, Arkadiy I. Garber, David Díaz Escandón, Andrew T. Cook, Spencer Goyette, Veera Tuovinen Nogerius, Tor Tønsberg, Lisa Y. Stein, Robert D. Finn, Toby Spribille.

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
