## [Editor Report · Decision Letter 0]

28 Feb 2024

Dear Dr Spribille, 

Thank you for submitting your manuscript entitled "Microbial occurrence and symbiont detection in lichen metagenomes" for consideration as a Research Article by PLOS Biology.

Your manuscript has now been evaluated by the PLOS Biology editorial staff, as well as by an academic editor with relevant expertise, and I'm writing to let you know that we would like to consider your paper further.

IMPORTANT: We're still deciding to what extent it needs further peer review; sorry not to be able to clarify that yet. Please ignore anything in the following letter template that relates to peer review, as the jury is still out...

However, before we can consider your manuscript further, we need you to complete your submission by providing the metadata that is required for full assessment. To this end, please login to Editorial Manager where you will find the paper in the 'Submissions Needing Revisions' folder on your homepage. Please click 'Revise Submission' from the Action Links and complete all additional questions in the submission questionnaire.

Once your full submission is complete, your paper will undergo a series of checks in preparation for peer review. After your manuscript has passed the checks it will be sent out for review. To provide the metadata for your submission, please Login to Editorial Manager (https://www.editorialmanager.com/pbiology) within two working days, i.e. by Mar 01 2024 11:59PM.

Kind regards,

Roli Roberts

Roland Roberts, PhD

Senior Editor

PLOS Biology

rroberts@plos.org

---

## [Decision Letter · Decision Letter 1]

15 May 2024

Dear Toby,

Thank you for your patience while your manuscript "Microbial occurrence and symbiont detection in lichen metagenomes" was peer-reviewed by PLOS Biology. Your work was assessed and discussed by the PLOS Biology editorial team, an Academic Editor with relevant expertise, and by three independent reviewers. As previously discussed, as part of our "portable peer review" initiative, all three reviewers were privy to the prior reviews from Nature and Cell Genomics. Based on the reviews, which you will find at the end of this email, I regret that we will not be pursuing your manuscript for publication in the journal.

You'll see that reviewer #1 is positive, saying that the paper is now more nuanced and re-framed in a more cautionary manner. Their requests are textual and fairly minor. Reviewer #2 starts positive but thinks that you are too dismissive of the possibility that contamination could have occurred during DNA isolation, and worries about your data being uploaded to GenBank without caveats; s/he also says that you missed an opportunity by not sequencing the substrate from which the lichens were obtained. Reviewer #3 thinks that the you still imply symbiosis too readily and is concerned that you propose to write a perspective piece about the topic. S/he recommends further re-framing the manuscript around issues that affect MAGs, rather than relationships between organisms; the rest of this review is a series of points about how your current manuscript is a halfway house, and how to make it more about the pros/cons of MAG approaches. I should say that two of the reviewers explicitly recommended that we reject the paper.

We solicited cross-comments from the reviewers, and discussed these and the reviews with the Academic Editor. Unfortunately the AE was not supportive of considering your study further at PLOS Biology, even in a much-caveated and tempered form.

I'm sorry that we cannot be more positive on this occasion, especially after this lengthy process, and hope the reviewer reports will help you in preparing your manuscript for submission elsewhere. We would be happy to transfer the peer reviews, along with the names of the reviewers (where they agree), to another journal of your choice. To start this process, we need an explicit request by the corresponding author.

I hope you understand the reasons for this decision and will consider PLOS Biology for other submissions in the future. Thank you for your support of PLOS and of Open Access publishing.

Sincerely,

Roli

Roland Roberts, PhD

Senior Editor

PLOS Biology

rroberts@plos.org

REVIEWERS' COMMENTS:

Reviewer #1:

Tagirdzhanova et al. 

Microbial occurrence and symbiont detection in lichen metagenomes

This manuscript is a resubmission of a manuscript previously reviewed at Nature and Cell Genomics. I was a previous reviewer and have read the extensively revised manuscript and the complex set of reviews and author responses from the earlier submissions. The authors have made a major effort to address my previous comments and, I believe, those of the other reviewers. I recommend that the paper be accepted with minor revision. 

Specific comments:

-Most generally, I want to highlight what I see as the most important element of the revised manuscript. It has a revised thrust and presents an important set of take-home messages and cautionary tales for lichen researchers, and metagenomicists in general. I won't rehash them here, but basically they are as follows: given what we know about the biology of lichens, there are things we can trust and things we cannot trust about MAG data. We now know enough to know that there are a variety of factors contribute to the extent to which key players in lichens are recovered in MAGs. Sometimes they are not recovered at all, despite knowing from other lines of inquiry that they are there (e.g., basidiomycetes). This is sobering and important for the community to take on board.

-Second, my opinion is that the authors have taken a suitably nuanced perspective of the concept of symbiosis and acknowledge the myriad uncertainties. The Discussion section on page 25-26 stands out in this regard, as does the "Limits to inference" section that immediately follows, which brings the question of 'surface sterilization' out into the open. The authors seem to be asking the question: do lichens [whatever that means] have a microbiome? I think this is an important question to ask and debate amongst researchers in the field. Side-stepping use of the term 'core' doesn't make the underlying issues go away. 

-The introduction is truly excellent - it provides an authoritative overview of the history and current state of affairs in modern lichen research. It reads a bit like a thesis intro (maybe it is); certainly the extensive referencing (well over 100 citations) is a strength of the paper, to the extent that it aligns with the constraints of PLOS Biology. 

-The 24 newly generated metagenomes are basically an afterthought in the current paper. Yes, they are embedded within a much larger study and analytical / interpretive framework, but readers will be interested in where they came from and how they were generated (provided on page 19, the very last section of the Results). Space permitting, draw more attention to it in the abstract (line 31) and consider moving this further up the results. I see why it is placed where it is - but maybe a nod to these data and their significance earlier would be helpful too. 

-I am not sure why the authors chose to embed the figure legends but not the figures into the main text. This was not helpful. 

-Line 127 - "We began by removing identical metagenomes that have been published twice in the sequence databases (n=43; S3 Table)". Meaning unclear based on main text alone. My brain instantly asked: do they mean literally the same dataset or independently sequenced metagenomes from the same location / environment by different labs? The former makes sense to remove, the latter would seem to be a useful control for experimental and bioinformatic procedures. This becomes clear from the supp info line 1381/82, i.e., they are literally the same data submitted twice to GenBank with different accession numbers, which of course is unfortunate. I suggest the authors make this clear in the main text as well. 

-For much of the intro, the LFS acronym is used in the singular (e.g., "the lichen fungal symbiont (LFS),..."). This of course becomes clear that they are indeed exploring and discussing the presence of basidiomycetes as well. I suggest working in acknowledgement of the presence of multiple types of fungi earlier on. Why? Even non-specialists / lichen enthusiasts will be aware of the high-profile Science paper describing multiple fungal partners. The way the intro reads at first it almost sounds as though the authors are back-tracking on that observation, which they are not.

Reviewer #2:

This manuscript presents an extensive analysis of lichen metagenomes aimed at elucidating the microbial composition and detecting symbiotic relationships within lichen ecosystems. The authors have reassembled and analyzed over 400 publicly available metagenomes, generating metagenome-assembled genomes (MAGs), constructing phylogenomic trees, and mapping MAG occurrence. The study provides valuable insights into the prevalence and diversity of bacterial, algal, and fungal constituents across a wide array of lichen species.

Before the submission to PLoS Biology, the manuscript was submitted to the journals Nature and Cell. Reviews from these earlier submissions were attached to this submission. 

The authors of the manuscript have made a significant change in the latest version by reducing the emphasis on describing a "core" and "global" set of microbial lichen symbionts. This change aligns with the recommendation of many reviewers. Although the authors' responses to the reviews suggest that they are not entirely convinced that their data does not allow such an evaluation, the changes are still appreciated. The new submission compares various methods to extract lichen metagenomes and presents the results more holistically by focusing on the total pool of genomes recoverable from the lichen data. 

Reviewers have pointed out that there is contamination in both the public dataset and the data produced by the authors. I disagree with the dismissive response from the authors. The contamination could have occurred during DNA isolation, before the barcoding. Although the authors provide an unlikely biological explanation, I am surprised they decided to ignore this contamination. I assume they still plan to upload this data without commenting on it to Genbank, which could cause problems for colleagues who want to work with the data. Furthermore, I am concerned that these samples may be representative of other inconsistencies in their data. 

Additionally, I believe there was a missed opportunity with the data they produced themselves. They could have addressed many of the criticisms raised by the reviewers by conducting further testing. For instance, they could have sequenced the substrate in addition to the lichen symbiosis to determine how many of the bacteria identified by sequencing were associated with the lichen and how many were sequenced by chance. Furthermore, they could have treated the same sample differently to test if they were correct in their speculation that irrespective of the treatment, the sequencing results would have remained the same.

The manuscript uses a metagenomic methodology to assess lichen symbioses, which is a valuable addition to the field. However, I have concerns regarding the inconsistencies in the presented data and the missed opportunities for further investigation. These factors cast doubt on the manuscript's suitability for publication in PLoS Biology.

Reviewer #3:

Dear Authors,

I have carefully reviewed the manuscript and the comments of other referees. Although I admire the effort behind putting together a manuscript, I have several major concerns, which prevent me from recommending its publication in its current form. Some of my major concerns align with those of the referees who reviewed the manuscript before.

Firstly, while the manuscript has shown improvement in certain areas, such as refraining from referring to the identified bacteria as core, the central narrative remains somewhat vague. Throughout the manuscript, you imply that bacteria are symbionts, yet refrain from stating this directly due to the lack of sufficient evidence to support the claim, as advised by most referees.

Specifically, while the referees have expressed concerns about the adequacy of evidence supporting this classification, you argue that the presence of bacteria, including on the unsterilised surface justifies them to be part of the biofilm and you also propose that bacterial gene pathways are beneficial and functionally relevant for fungi based on their mere presence in bacteria. However, the sole presence of microorganisms on an unsterilized surface does not necessarily imply a symbiotic relationship. Do we consider viruses and contaminants from the environment, such as dust and sand particles, as a part of biofilm and tag them as symbionts? No. Then why bacteria. Simply because bacteria have the pathways that could potentially be beneficial for fungus does not mean the identified bacteria are symbiont. Do closely related non-symbiont bacteria lack these pathways?

Until sufficient evidence is generated, I would suggest to refrain from making such claims.

Furthermore, your reliance on publishing a perspective to support your argument may indicate a lack of openness to alternative viewpoints, which undermines the purpose of peer review. You have faint to no support for their idea of bacteria being symbionts but you argue that you will publish a perspective on this. So you will increase the support for your claim by publishing opinions and perspectives. We should be careful about supporting such approach that an idea/perspective is suggested as a hypothesis but becomes a "suggestion" and eventually a fact. A perspective therefore would not give more support to bacteria being symbionts.

Moreover, I suggest refraining from name-calling, such as "Duke school," and instead showing respect for alternative hypotheses. I understand the frustration given the work put into a manuscript. However, given the insufficient evidence to support a strong symbiotic relationship, it's important to acknowledge and address the specific concerns raised by the referees.

As the central idea of the manuscript is not supported by the evidence, I would recommend major restructuring.

Best regards.

Major concerns:

1) The manuscript introduction appears to lack a clear direction, with an imbalance evident in all the sections. For instance, given the title of the manuscript "Microbial occurrence and symbiont detection in lichen metagenomes" and the main knowledge gap being "overall content and limitations of these metagenomes have not been assessed," I would expect the introduction to center around issues of MAGs and risks of bacterial read contamination. Instead, there is an undue emphasis on explaining the relationship between fungi and algae. This suggests that the manuscript may have undergone several rewrites, leading to a loss of focus and coherence in conveying the central story.

2) My second major concern aligns with one of the reviewers, regarding the use of MAGs to identify lichen-associated bacteria. This also explains the use of convoluted terminology between the abstract and introduction to state the main aim of the paper. While the manuscript progresses, the authors shift from "overall content and limitations of MAGs have not been assessed" to implying consistently that bacteria are symbionts but without ever stating it directly. They keep referring to the studies that investigated the possibility that bacteria are symbionts.

3) Between the two possible directions of the manuscript (bacteria in MAG or bacteria as lichen symbionts), based on the data, I would recommend opting for the first one as hinted in the abstract. For the second option, 1) the data is not apt, and 2) even if the presence is established, there is not enough or any evidence for stating that bacteria are symbionts (see detailed comments below).

4) My fourth but actually one of the biggest concerns is that despite various reviewers in two rounds of reviews, the authors are not keeping an open mind to the fact that several unrelated reviewers, for one reason or another, but all very strong, advise the authors that they do not have evidence to even suggest that bacteria are symbiotic to lichen associations or lichens are composed of fungus, algae, and bacteria. It seems that the authors got fixed on the idea of proving bacteria are symbiotic in lichens, and they are looking at everything through that lens, completely ignoring the concerns of all the reviewers.

5) Misleading idea. Starting with bacteria in MAGs to slowly transitioning to bacteria are symbionts is conveying that authors want to make a claim without evidence but being stopped by the reviewers' refrain to state it directly but keep implying it through the manuscript. I would strongly recommend against it. Please define an aim, state a knowledge gap, address it, and REFRAIN not only from stating but also implying the conclusions not supported by evidence.

Specific comments

1) The manuscript introduction appears to be lacking a clear direction, with an imbalance evident in all the sections. 

For instance given the title of the manuscript "Microbial occurrence and symbiont detection in lichen metagenomes", and the main knowledge gap being "overall content and limitations of these metagenomes have not been assessed" I would expect the introduction centers around issues of MAGs and risks of bacterial read contamination.

Instead, there is an undue emphasis on explaining the relationship between fungi and algae. This suggests that the manuscript may have undergone several rewrites, leading to a loss of focus and coherence in conveying the central story.

As the introduction flows the primary focus of the paper, i.e., bacteria in MAG (if we read the title and abstract), shifts to bacteria as lichen symbionts. See below

Abstract: In lichen research, metagenomes are increasingly being used for evaluating symbiont composition and metabolic potential, but the overall content and limitations of these metagenomes have not been assessed. We reassembled over 400 publicly available metagenomes, generated metagenome-assembled genomes (MAGs), constructed phylogenomic trees and mapped MAG occurrence and frequency across the dataset. 

Introduction: (lines 86-89) However, though hundreds of bacterial strains have been detected, overall few lichen symbioses have been censused for their bacterial composition, with most work focused on the model symbiosis involving the ascomycete fungus Lobaria pulmonaria (reviewed by Grimm et al. [13]). 

Discussion:

lines 446-450

Our analyses of lichen metagenomes yielded two contrasting sets of results. On the one hand, our data provide evidence that bacteria that have until now been studied as potential lichen symbionts in only a couple dozen lichens in fact occur in or on hundreds of lichens, and by extracting over 600 MAGs we increase the amount of available genomic data by orders of magnitude. 

Results:

If the paper is about overall content and limitations of these MAGs

What is the logic of analyzing the bacterial pathways and genes and their functional categories? In that case authors analyze the pathways present in bacteria from different labs when the samples were prepared

If the paper analyzes the possibility of bacteria being symbionts 

1) The dataset is not suitable. In short most lichen genomes are generated to obtain the fungus and hence may not implement appropriate sample preparation protocol and environment. 

2) As the data is generated by several different labs, the amount of bacteria in the samples is uncomparable.

3) Even if we ignore these points and consider the dataset is suitable for the study, the study only shows bacteria are present in the thalli but if the pathways that bacteria contain benefit the fungus is anything but pure supposition, especially because some of these bacteria may also be free living or associated with other organisms. In that case one must prove that these pathways exist entirely in bacteria symbiotic with lichens to say that these pathways are relevant for fungi and hence bacteria are symbionts.

I would suggest to define the aim of the manuscript clearly it can either be :

Bacterial contamination in lichenized fungal MAGs; As evident from the abstract lines 28-32

In lichen research, metagenomes are increasingly being used for evaluating symbiont composition and metabolic potential, but the overall content and limitations of these metagenomes have not been assessed. We reassembled over 400 publicly available metagenomes, generated metagenome-assembled genomes (MAGs), constructed phylogenomic trees and mapped MAG occurrence and frequency across the dataset."

Or 

Screening lichens for the bacterial composition/symbionts; As implied in the introduction lines 86-89

However, though hundreds of bacterial strains have been detected, overall few lichen symbioses have been censused for their bacterial composition, with most work focused on the model symbiosis involving the ascomycete fungus Lobaria pulmonaria (reviewed by Grimm et al. [13]). 

But not implying bacteria as symbionts in either of the above

Introduction contains details and knowledge gaps which have nothing to do with this title or abstract of the article 

for instance:

73-76 Although much about the biology of the disparate photobiont lineages remains unclear, especially with respect to what they receive from the fungus, their population structuring vis-a-vis their LFS partners is consistent with the definition of an open symbiosis [20]. 

Another example: No consensus exists however on how many other symbionts might be involved, and since most other associated microbes lack visible chlorophyll, their detection is not trivial. Researchers as early as Maria Cengia Sambo in the 1920s cultured additional bacteria from lichens, leading to the speculation that lichens could be 'polysymbioses' of more than two partners.

This is an example of slowly shifting focus from bacteria in MAG (implying that bacteria are symbionts?)

The major section of results deals with the functional profile of the detected bacteria (line 311 Functional profiles of the most commonly detected bacteria, line 342-367 Carbon usage, 368-377 Aerobic an oxygenic phototrophy, 378-388 Nitrogen fixation, 389-410 Cofactor and iron scavenging, 411-426 Carbohydrate degradation). But unless there is a proof that bacteria are symbiotic to lichens we are reading about which pathways do bacteria detected on surface of lichens have. 

Discussion 

Major part of discussion (lines 446-525) is focussed on proving why bacteria and algae may go undetected in lichens to support that bacteria are common occurrence (=symbiotic according to authors). They refer to studies which have and have not found Cystobasidiomycetes, photobionts etc. in lichens before. Regarding this, first this is of methodological relevance and does not require a major section in discussion and second still it only shows that bacteria are present.

---

## [Editor Report · Decision Letter 2]

6 Jun 2024

Dear Toby,

Many thanks for your patience while we considered your Appeal of our previous decision on your manuscript "Microbial occurrence and symbiont detection in lichen metagenomes." As mentioned in my previous email, after discussion among the team and with the Academic Editor, we think that despite the concerns raised, there is significant value in your study, and have decided to invite you to revise your manuscript, taking measures to allay the reviewers' substantial and understandable concerns.

As previously discussed, reviewer #1 is positive, saying that the paper is now more nuanced and re-framed in a more cautionary manner. Their requests are textual and fairly minor. Reviewer #2 starts positive but thinks that you are too dismissive of the possibility that contamination could have occurred during DNA isolation, and worries about your data being uploaded to GenBank without caveats; s/he also says that you missed an opportunity by not sequencing the substrate from which the lichens were obtained. Reviewer #3 thinks that the you still imply symbiosis too readily and is concerned that you propose to write a perspective piece about the topic. S/he recommends further re-framing the manuscript around issues that affect MAGs, rather than relationships between organisms; the rest of this review is a series of points about how your current manuscript is a halfway house, and how to make it more about the pros/cons of MAG approaches.

Your Appeal email rebutted some of the points raised by reviewers #2 and #3, but also proposed constructive measure to address their remaining concerns. As I understand it, these include a) being completely explicit about having zero direct evidence for symbiosis, b) making a circumstantial but strong case for the likelihood of contamination being negligible, c) clarifying why uploading the MAGs is “industry standard” and should not be controversial or problematic, d) incorporating text about the “open” nature of well accepted lichen symbioses (and why this reduces the onus on analysing substrate metagenomes), and e) including the eukaryotic side of the KEGG/pathway analysis to complement the current bacterial ones presented in Fig 6. To my mind, a reader should not walk away from this paper thinking that you have unequivocally demonstrated a symbiotic role (sensu stricto) for the bacterial components, but that the very useful data have been presented even-handedly and the circumstantial nature of the evidence for symbiosis made clear.

I should also say that when discussing the Appeal decision, the Academic Editor added the following, which you might find helpful: "One clarification that would help prevent misunderstandings is a clear definition of how the term 'symbiont' is being used. The original definition was just organisms of different species in sustained, close physical contact. But a common use of the term is more specific -- an organism playing some role in the interaction. And an even more specific definition is an organism playing a positive role by enhancing fitness of other participants (i.e. mutualists, not parasites). There is clear evidence for the first meaning, not so clear for the latter meanings."

In light of the reviews, which you will find at the end of this email, we would like to invite you to revise the work to thoroughly address the reviewers' reports.

Given the extent of revision needed, we cannot make a decision about publication until we have seen the revised manuscript and your response to the reviewers' comments. Your revised manuscript is likely to be sent for further evaluation by all or a subset of the reviewers.

**IMPORTANT - SUBMITTING YOUR REVISION**

*Re-submission Checklist*

*Published Peer Review*

*PLOS Data Policy*

*Blot and Gel Data Policy*

Sincerely,

Roli

Roland Roberts, PhD

Senior Editor

PLOS Biology

rroberts@plos.org

REVIEWERS' COMMENTS:

Reviewer #1:

Tagirdzhanova et al.

Microbial occurrence and symbiont detection in lichen metagenomes

This manuscript is a resubmission of a manuscript previously reviewed at Nature and Cell Genomics. I was a previous reviewer and have read the extensively revised manuscript and the complex set of reviews and author responses from the earlier submissions. The authors have made a major effort to address my previous comments and, I believe, those of the other reviewers. I recommend that the paper be accepted with minor revision.

Specific comments:

-Most generally, I want to highlight what I see as the most important element of the revised manuscript. It has a revised thrust and presents an important set of take-home messages and cautionary tales for lichen researchers, and metagenomicists in general. I won't rehash them here, but basically they are as follows: given what we know about the biology of lichens, there are things we can trust and things we cannot trust about MAG data. We now know enough to know that there are a variety of factors contribute to the extent to which key players in lichens are recovered in MAGs. Sometimes they are not recovered at all, despite knowing from other lines of inquiry that they are there (e.g., basidiomycetes). This is sobering and important for the community to take on board.

-Second, my opinion is that the authors have taken a suitably nuanced perspective of the concept of symbiosis and acknowledge the myriad uncertainties. The Discussion section on page 25-26 stands out in this regard, as does the "Limits to inference" section that immediately follows, which brings the question of 'surface sterilization' out into the open. The authors seem to be asking the question: do lichens [whatever that means] have a microbiome? I think this is an important question to ask and debate amongst researchers in the field. Side-stepping use of the term 'core' doesn't make the underlying issues go away.

-The introduction is truly excellent - it provides an authoritative overview of the history and current state of affairs in modern lichen research. It reads a bit like a thesis intro (maybe it is); certainly the extensive referencing (well over 100 citations) is a strength of the paper, to the extent that it aligns with the constraints of PLOS Biology.

-The 24 newly generated metagenomes are basically an afterthought in the current paper. Yes, they are embedded within a much larger study and analytical / interpretive framework, but readers will be interested in where they came from and how they were generated (provided on page 19, the very last section of the Results). Space permitting, draw more attention to it in the abstract (line 31) and consider moving this further up the results. I see why it is placed where it is - but maybe a nod to these data and their significance earlier would be helpful too.

-I am not sure why the authors chose to embed the figure legends but not the figures into the main text. This was not helpful.

-Line 127 - "We began by removing identical metagenomes that have been published twice in the sequence databases (n=43; S3 Table)". Meaning unclear based on main text alone. My brain instantly asked: do they mean literally the same dataset or independently sequenced metagenomes from the same location / environment by different labs? The former makes sense to remove, the latter would seem to be a useful control for experimental and bioinformatic procedures. This becomes clear from the supp info line 1381/82, i.e., they are literally the same data submitted twice to GenBank with different accession numbers, which of course is unfortunate. I suggest the authors make this clear in the main text as well.

-For much of the intro, the LFS acronym is used in the singular (e.g., "the lichen fungal symbiont (LFS),..."). This of course becomes clear that they are indeed exploring and discussing the presence of basidiomycetes as well. I suggest working in acknowledgement of the presence of multiple types of fungi earlier on. Why? Even non-specialists / lichen enthusiasts will be aware of the high-profile Science paper describing multiple fungal partners. The way the intro reads at first it almost sounds as though the authors are back-tracking on that observation, which they are not.

Reviewer #2:

This manuscript presents an extensive analysis of lichen metagenomes aimed at elucidating the microbial composition and detecting symbiotic relationships within lichen ecosystems. The authors have reassembled and analyzed over 400 publicly available metagenomes, generating metagenome-assembled genomes (MAGs), constructing phylogenomic trees, and mapping MAG occurrence. The study provides valuable insights into the prevalence and diversity of bacterial, algal, and fungal constituents across a wide array of lichen species.

Before the submission to PLoS Biology, the manuscript was submitted to the journals Nature and Cell. Reviews from these earlier submissions were attached to this submission.

The authors of the manuscript have made a significant change in the latest version by reducing the emphasis on describing a "core" and "global" set of microbial lichen symbionts. This change aligns with the recommendation of many reviewers. Although the authors' responses to the reviews suggest that they are not entirely convinced that their data does not allow such an evaluation, the changes are still appreciated. The new submission compares various methods to extract lichen metagenomes and presents the results more holistically by focusing on the total pool of genomes recoverable from the lichen data.

Reviewers have pointed out that there is contamination in both the public dataset and the data produced by the authors. I disagree with the dismissive response from the authors. The contamination could have occurred during DNA isolation, before the barcoding. Although the authors provide an unlikely biological explanation, I am surprised they decided to ignore this contamination. I assume they still plan to upload this data without commenting on it to Genbank, which could cause problems for colleagues who want to work with the data. Furthermore, I am concerned that these samples may be representative of other inconsistencies in their data.

Additionally, I believe there was a missed opportunity with the data they produced themselves. They could have addressed many of the criticisms raised by the reviewers by conducting further testing. For instance, they could have sequenced the substrate in addition to the lichen symbiosis to determine how many of the bacteria identified by sequencing were associated with the lichen and how many were sequenced by chance. Furthermore, they could have treated the same sample differently to test if they were correct in their speculation that irrespective of the treatment, the sequencing results would have remained the same.

The manuscript uses a metagenomic methodology to assess lichen symbioses, which is a valuable addition to the field. However, I have concerns regarding the inconsistencies in the presented data and the missed opportunities for further investigation. These factors cast doubt on the manuscript's suitability for publication in PLoS Biology.

Reviewer #3:

Dear Authors,

I have carefully reviewed the manuscript and the comments of other referees. Although I admire the effort behind putting together a manuscript, I have several major concerns, which prevent me from recommending its publication in its current form. Some of my major concerns align with those of the referees who reviewed the manuscript before.

Firstly, while the manuscript has shown improvement in certain areas, such as refraining from referring to the identified bacteria as core, the central narrative remains somewhat vague. Throughout the manuscript, you imply that bacteria are symbionts, yet refrain from stating this directly due to the lack of sufficient evidence to support the claim, as advised by most referees.

Specifically, while the referees have expressed concerns about the adequacy of evidence supporting this classification, you argue that the presence of bacteria, including on the unsterilised surface justifies them to be part of the biofilm and you also propose that bacterial gene pathways are beneficial and functionally relevant for fungi based on their mere presence in bacteria. However, the sole presence of microorganisms on an unsterilized surface does not necessarily imply a symbiotic relationship. Do we consider viruses and contaminants from the environment, such as dust and sand particles, as a part of biofilm and tag them as symbionts? No. Then why bacteria. Simply because bacteria have the pathways that could potentially be beneficial for fungus does not mean the identified bacteria are symbiont. Do closely related non-symbiont bacteria lack these pathways?

Until sufficient evidence is generated, I would suggest to refrain from making such claims.

Furthermore, your reliance on publishing a perspective to support your argument may indicate a lack of openness to alternative viewpoints, which undermines the purpose of peer review. You have faint to no support for their idea of bacteria being symbionts but you argue that you will publish a perspective on this. So you will increase the support for your claim by publishing opinions and perspectives. We should be careful about supporting such approach that an idea/perspective is suggested as a hypothesis but becomes a "suggestion" and eventually a fact. A perspective therefore would not give more support to bacteria being symbionts.

Moreover, I suggest refraining from name-calling, such as "Duke school," and instead showing respect for alternative hypotheses. I understand the frustration given the work put into a manuscript. However, given the insufficient evidence to support a strong symbiotic relationship, it's important to acknowledge and address the specific concerns raised by the referees.

As the central idea of the manuscript is not supported by the evidence, I would recommend major restructuring.

Best regards.

Major concerns:

1) The manuscript introduction appears to lack a clear direction, with an imbalance evident in all the sections. For instance, given the title of the manuscript "Microbial occurrence and symbiont detection in lichen metagenomes" and the main knowledge gap being "overall content and limitations of these metagenomes have not been assessed," I would expect the introduction to center around issues of MAGs and risks of bacterial read contamination. Instead, there is an undue emphasis on explaining the relationship between fungi and algae. This suggests that the manuscript may have undergone several rewrites, leading to a loss of focus and coherence in conveying the central story.

2) My second major concern aligns with one of the reviewers, regarding the use of MAGs to identify lichen-associated bacteria. This also explains the use of convoluted terminology between the abstract and introduction to state the main aim of the paper. While the manuscript progresses, the authors shift from "overall content and limitations of MAGs have not been assessed" to implying consistently that bacteria are symbionts but without ever stating it directly. They keep referring to the studies that investigated the possibility that bacteria are symbionts.

3) Between the two possible directions of the manuscript (bacteria in MAG or bacteria as lichen symbionts), based on the data, I would recommend opting for the first one as hinted in the abstract. For the second option, 1) the data is not apt, and 2) even if the presence is established, there is not enough or any evidence for stating that bacteria are symbionts (see detailed comments below).

4) My fourth but actually one of the biggest concerns is that despite various reviewers in two rounds of reviews, the authors are not keeping an open mind to the fact that several unrelated reviewers, for one reason or another, but all very strong, advise the authors that they do not have evidence to even suggest that bacteria are symbiotic to lichen associations or lichens are composed of fungus, algae, and bacteria. It seems that the authors got fixed on the idea of proving bacteria are symbiotic in lichens, and they are looking at everything through that lens, completely ignoring the concerns of all the reviewers.

5) Misleading idea. Starting with bacteria in MAGs to slowly transitioning to bacteria are symbionts is conveying that authors want to make a claim without evidence but being stopped by the reviewers' refrain to state it directly but keep implying it through the manuscript. I would strongly recommend against it. Please define an aim, state a knowledge gap, address it, and REFRAIN not only from stating but also implying the conclusions not supported by evidence.

Specific comments

1) The manuscript introduction appears to be lacking a clear direction, with an imbalance evident in all the sections.

For instance given the title of the manuscript "Microbial occurrence and symbiont detection in lichen metagenomes", and the main knowledge gap being "overall content and limitations of these metagenomes have not been assessed" I would expect the introduction centers around issues of MAGs and risks of bacterial read contamination.

Instead, there is an undue emphasis on explaining the relationship between fungi and algae. This suggests that the manuscript may have undergone several rewrites, leading to a loss of focus and coherence in conveying the central story.

As the introduction flows the primary focus of the paper, i.e., bacteria in MAG (if we read the title and abstract), shifts to bacteria as lichen symbionts. See below

Abstract: In lichen research, metagenomes are increasingly being used for evaluating symbiont composition and metabolic potential, but the overall content and limitations of these metagenomes have not been assessed. We reassembled over 400 publicly available metagenomes, generated metagenome-assembled genomes (MAGs), constructed phylogenomic trees and mapped MAG occurrence and frequency across the dataset.

Introduction: (lines 86-89) However, though hundreds of bacterial strains have been detected, overall few lichen symbioses have been censused for their bacterial composition, with most work focused on the model symbiosis involving the ascomycete fungus Lobaria pulmonaria (reviewed by Grimm et al. [13]).

Discussion:

lines 446-450

Our analyses of lichen metagenomes yielded two contrasting sets of results. On the one hand, our data provide evidence that bacteria that have until now been studied as potential lichen symbionts in only a couple dozen lichens in fact occur in or on hundreds of lichens, and by extracting over 600 MAGs we increase the amount of available genomic data by orders of magnitude.

Results:

If the paper is about overall content and limitations of these MAGs

What is the logic of analyzing the bacterial pathways and genes and their functional categories? In that case authors analyze the pathways present in bacteria from different labs when the samples were prepared

If the paper analyzes the possibility of bacteria being symbionts

1) The dataset is not suitable. In short most lichen genomes are generated to obtain the fungus and hence may not implement appropriate sample preparation protocol and environment.

2) As the data is generated by several different labs, the amount of bacteria in the samples is uncomparable.

3) Even if we ignore these points and consider the dataset is suitable for the study, the study only shows bacteria are present in the thalli but if the pathways that bacteria contain benefit the fungus is anything but pure supposition, especially because some of these bacteria may also be free living or associated with other organisms. In that case one must prove that these pathways exist entirely in bacteria symbiotic with lichens to say that these pathways are relevant for fungi and hence bacteria are symbionts.

I would suggest to define the aim of the manuscript clearly it can either be :

Bacterial contamination in lichenized fungal MAGs; As evident from the abstract lines 28-32

In lichen research, metagenomes are increasingly being used for evaluating symbiont composition and metabolic potential, but the overall content and limitations of these metagenomes have not been assessed. We reassembled over 400 publicly available metagenomes, generated metagenome-assembled genomes (MAGs), constructed phylogenomic trees and mapped MAG occurrence and frequency across the dataset."

Or

Screening lichens for the bacterial composition/symbionts; As implied in the introduction lines 86-89

However, though hundreds of bacterial strains have been detected, overall few lichen symbioses have been censused for their bacterial composition, with most work focused on the model symbiosis involving the ascomycete fungus Lobaria pulmonaria (reviewed by Grimm et al. [13]).

But not implying bacteria as symbionts in either of the above

Introduction contains details and knowledge gaps which have nothing to do with this title or abstract of the article

for instance:

73-76 Although much about the biology of the disparate photobiont lineages remains unclear, especially with respect to what they receive from the fungus, their population structuring vis-a-vis their LFS partners is consistent with the definition of an open symbiosis [20].

Another example: No consensus exists however on how many other symbionts might be involved, and since most other associated microbes lack visible chlorophyll, their detection is not trivial. Researchers as early as Maria Cengia Sambo in the 1920s cultured additional bacteria from lichens, leading to the speculation that lichens could be 'polysymbioses' of more than two partners.

This is an example of slowly shifting focus from bacteria in MAG (implying that bacteria are symbionts?)

The major section of results deals with the functional profile of the detected bacteria (line 311 Functional profiles of the most commonly detected bacteria, line 342-367 Carbon usage, 368-377 Aerobic an oxygenic phototrophy, 378-388 Nitrogen fixation, 389-410 Cofactor and iron scavenging, 411-426 Carbohydrate degradation). But unless there is a proof that bacteria are symbiotic to lichens we are reading about which pathways do bacteria detected on surface of lichens have.

Discussion

Major part of discussion (lines 446-525) is focussed on proving why bacteria and algae may go undetected in lichens to support that bacteria are common occurrence (=symbiotic according to authors). They refer to studies which have and have not found Cystobasidiomycetes, photobionts etc. in lichens before. Regarding this, first this is of methodological relevance and does not require a major section in discussion and second still it only shows that bacteria are present.

---

## [Editor Report · Decision Letter 3]

11 Sep 2024

Dear Dr Spribille,

Thank you for your patience while we considered your revised manuscript "Microbial occurrence and symbiont detection in lichen metagenomes" for publication as a Research Article at PLOS Biology. This revised version of your manuscript has been evaluated by the PLOS Biology editors, the Academic Editor.

Based oon our Academic Editor's assessment of your revision, we are likely to accept this manuscript for publication, provided you satisfactorily address the remaining editorial points. Please also make sure to address the following data and other policy-related requests.

a) We routinely suggest changes to titles to ensure maximum accessibility for a broad, non-specialist readership, and to ensure they reflect the contents of the paper. In this case, we would suggest a minor edit to the title, as follows. Please ensure you change both the manuscript file and the online submission system, as they need to match for final acceptance:

"Microbial occurrence and symbiont detection in global lichen metagenomes"

b) Thank you for providing the funding institutions. Please also provide the grant number for each grant.

c) We have no word limit. Please place the "extended" methods and results in the main text.

Please supply the numerical values either in the a supplementary file or as a permanent DOI’d deposition for the following figures:

Figure 4AB, 5ABC, 6, S1, S4, S6, S7

e) Please cite the location of the data clearly in all relevant main and supplementary Figure legends, e.g. “The data underlying this Figure can be found in S1 Data” or “The data underlying this Figure can be found in https://doi.org/10.5281/zenodo.XXXXX”

f) Please also provide the tree files for Figures 2ABC, S5

g) I notice the statement saying “All MAGs, assemblies, and de novo generated raw data are submitted to ENA under the study accession PRJEB59037 (data release pending). Phylogenomic trees in Newick format are available at FigShare (pending). Custom scripts used for data analysis and visualization are available on GitHub (https://github.com/Spribille-lab/Aglobal-survey-of-lichen-symbionts-from-metagenomes) and FigShare (pending).”. We will require access before accepting the manuscript.

h) Please ensure that your Data Statement in the submission system accurately describes where your data can be found and is in final format, as it will be published as written there.

i) Many thanks for providing the underlying code in GitHub. However, because Github depositions can be readily changed or deleted, please make a permanent DOI’d copy (e.g. in Zenodo) and provide this URL in the manuscript and Data Availability Statement.

We expect to receive your revised manuscript within two weeks. 

*Published Peer Review History*

*Press*

Sincerely,

Melissa

Melissa Vázquez Hernández, PhD

Associate Editor, PLOS Biology

on behalf of

Roland

Roland Roberts, PhD

Senior Editor

rroberts@plos.org

PLOS Biology

---

## [Editor Report · Decision Letter 4]

24 Sep 2024

Dear Toby,

Thank you for the submission of your revised Research Article "Microbial occurrence and symbiont detection in a global sample of lichen metagenomes" for publication in PLOS Biology. On behalf of my colleagues and the Academic Editor, Nancy Moran, I'm pleased to say that we can in principle accept your manuscript for publication, provided you address any remaining formatting and reporting issues. These will be detailed in an email you should receive within 2-3 business days from our colleagues in the journal operations team; no action is required from you until then. Please note that we will not be able to formally accept your manuscript and schedule it for publication until you have completed any requested changes.

Sincerely, 

Roli

Senior Editor

PLOS Biology

rroberts@plos.org